# Single-cell RNA sequencing of mouse islets exposed to proinflammatory cytokines

Jennifer S Stancill[1] , Moujtaba Y Kasmani[2,3] , Achia Khatun[2,3], Weiguo Cui[2,3], John A Corbett[1]

**Exposure to proinflammatory cytokines is believed to contribute to pancreatic β-cell damage during diabetes development. Although some cytokine-mediated changes in islet gene expression are known, the heterogeneity of the response is not well-understood. After 6-h treatment with IL-1β and IFN-γ alone or together, mouse islets were subjected to single-cell RNA sequencing. Treatment with both cytokines together led to expression of inducible nitric oxide synthase mRNA (*Nos2*) and antiviral and immune-associated genes in a subset of β-cells. Interestingly, IL-1β alone activated antiviral genes. Subsets of δ- and α-cells expressed *Nos2* and exhibited similar gene expression changes as β-cells, including increased expression of antiviral genes and repression of identity genes. Finally, cytokine responsiveness was inversely correlated with expression of genes encoding heat shock proteins. Our findings show that all islet endocrine cell types respond to cytokines, IL-1β induces the expression of protective genes, and cellular stress gene expression is associated with inhibition of cytokine signaling.**

## Introduction

Pancreatic islets of Langerhans are highly vascularized micro-organs containing heterogeneous populations of hormone-secreting endocrine cells that are responsible for controlling whole-body glucose homeostasis. The most abundant endocrine cell type is the insulin-secreting β-cell, which makes up about 70% of the rodent islet. Other endocrine cells include glucagon-secreting α-cells and somatostatin-secreting δ-cells, which comprise ~15% and 5% of the islet, respectively. There are also non-endocrine cells in the islets, including endothelial cells and resident macrophages (Bonner-Weir & Orci, 1982; Setum et al, 1991). Type 1 diabetes is an autoimmune disease caused by selective destruction of β-cells (Gepts, 1965) that leads to insulin deficiency and severe hyperglycemia if not treated by daily insulin injections. Whereas the killing of these cells is primarily mediated by

T-cell–dependent mechanisms (Like et al, 1985; Wicker et al, 1986; Bergman & Haskins, 1997), inflammatory cytokines such as IL-1β and IFN-γ produced by macrophages and T lymphocytes (Mandrup-Poulsen et al, 1985; Wright et al, 1988; Wright & Lacy, 1989) are also believed to contribute to β-cell damage and disease development. The role of cytokines in the disease process is based on the ability of these soluble mediators to impair β-cell function and induce β-cell death following ex vivo treatment. For example, IL-1β inhibits oxidative phosphorylation and insulin secretion while also inducing ER stress and DNA damage (Padgett et al, 2013; Eizirik et al, 2020). Nitric oxide, produced in β-cells after the expression of inducible nitric oxide synthase (iNOS) in response to IL-1β exposure, mediates these damaging effects (Southern et al, 1990; Corbett et al, 1991, 1992a, 1993; Welsh et al, 1991; Corbett & McDaniel, 1992). However, whereas IL-1β and nitric oxide are primarily viewed as damaging to the islet, we have recently shown that nitric oxide protects β-cells against DNA damage–induced apoptosis and against viral infection (Oleson et al, 2016; Stafford et al, 2020).

Cytokines have been shown to activate multiple pathways in β-cells. IL-1β stimulates nuclear factor kappa-light-chain-enhancer of activated B cells (NF-κB) and MAPK signaling (Eizirik et al, 1996; Larsen et al, 1998). NF-κB is required for the expression of inflammatory genes such as *Nos2* (Saldeen & Welsh, 1994; Kwon et al, 1995; Flodstrom et al, 1996; Darville & Eizirik, 1998). IFN-γ primarily activates JAK-STAT pathways as well as interferon regulatory factor (IRF)-1 expression (Mamane et al, 1999; Schneider et al, 2014). Interestingly, many inflammatory genes, including *Nos2*, contain IFN response elements in their promoters (Xie et al, 1993), perhaps explaining why IFN-γ potentiates their expression in β-cells (Heitmeier et al, 1997). IFN-γ is also responsible for the induction of many antiviral response genes that function to limit viral replication (Samuel, 2001).

In addition to *Nos2*, cytokines are known to induce robust islet gene expression changes, many of which are nitric oxide dependent (Cardozo et al, 2001b; Ylipaasto et al, 2005; Ortis et al, 2010; Hughes et al, 2011; Meares et al, 2013). However, previous studies have been limited by the use of whole islets, which contain several different cell types (Ylipaasto et al, 2005; Meares et al, 2013). Although β-cells make up most of the islet, it may not be accurate to assume that

[1]Department of Biochemistry, Medical College of Wisconsin, Milwaukee, WI, USA    [2]Department of Microbiology and Immunology, Medical College of Wisconsin, Milwaukee, WI, USA    [3]Blood Research Institute, Versiti, Milwaukee, WI, USA

Correspondence: jcorbett@mcw.edu; jstancill@mcw.edu

expression changes observed in islets can be attributed solely to changes in β-cells. This limitation can partially be overcome by using FACS-purified β-cells, which has been performed by several groups (Eizirik et al, 1992; Corbett et al, 1992b; Strandell et al, 1995; Cardozo et al, 2001a, 2001b), but this purification still ignores the heterogeneity that exists within the β-cell population (Pipeleers et al, 1994; Bader et al, 2016; Dorrell et al, 2016). In addition, the responses of non–β-cells within the islet to cytokines have largely been ignored, potentially because of the naturally low cell numbers in which these populations exist.

Here, we used single-cell RNA sequencing (scRNA-seq) to address these limitations. After exposing isolated mouse islets for 6 h to IL-1β or IFN-γ alone or in combination, we captured and sequenced transcriptomes from more than 7,700 single cells. Our analysis revealed heterogeneity in the β-cell population, with only 70% of β-cells increasing Nos2 in response to treatment with both cytokines. Cytokine nonresponsive cells were enriched for heat shock proteins and other chaperones, suggesting induction of a stress response. To our surprise, we observed that cytokines increase Nos2 expression in most of the δ-cells and in some of the α-cells, and that the response to cytokines is largely the same among all islet endocrine cell types. These responses include increase in antiviral and other immune response genes as well as repression of endocrine cell identity genes. Importantly, expression of selected genes was confirmed by qRT-PCR to be nitric oxide-independent. Our results suggest that islet responses to cytokines are not unique to β-cells but are similar throughout the entire endocrine population of the islet and are characterized by the stimulation of protective gene expression. Taken together with observations that nitric oxide protects β-cells from apoptosis and viral infection, the studies described here support a model in which the primary functions of cytokines are to protect islet endocrine cells from damage.

## Results

### scRNA-seq of mouse islets after 6-h cytokine exposure reveals cellular heterogeneity

Although some cytokine-mediated changes in islet gene expression are known, and several of the genes induced are protective, we do not have a complete picture of the cell type intrinsic changes stimulated by cytokines. To address this question, scRNA-seq was performed on untreated mouse islets (Sample 1) or mouse islets treated for 6 h with 10 U/ml IL-1β alone (Sample 2), 150 U/ml IFN-γ alone (Sample 3), or the combination of the two cytokines (Sample 4) (Fig 1A). After 6-h exposure, the levels of nitric oxide produced are low, such that observed gene expression changes are primarily nitric oxide-independent (Corbett et al, 1992a). After quality control, we recovered about 2,000 cells per sample. Cells from all samples were visualized using Uniform Manifold Approximation and Projection (UMAP), an algorithm that unbiasedly grouped the cells into 17 clusters based on similarity of gene expression (Fig 1B and Tables S1 and S2). We assigned endocrine cell identities (β-, α-, δ-, and PP-cells) based on enrichment of the four primary islet hormones

(insulin, glucagon, somatostatin, and pancreatic polypeptide, respectively) (Figs 1B and S1). β-cells (Ins1, Ins2) comprised 70% (5,436 cells) of our dataset, α-cells (Gcg) 12% (921 cells), δ-cells (Sst) 8% (651 cells), and PP-cells (Ppy) 2% (159 cells). Our analysis also identified two clusters (6 and 11, made up of β- and non-β endocrine cells, respectively) significantly enriched for geminin (Gmnn), an inhibitor of DNA replication that accumulates during the S phase of mitosis, suggesting that they may represent proliferative cells. Using characteristic gene expression, we identified the cell types of the non-endocrine clusters, which each make up less than 2% of our dataset (Figs 1B and S1): endothelial cells (Cd34), macrophages (Ccr5), mesenchymal cells (Col1a1), ductal cells (Krt17), and acinar cells (Cpa1). Notably, Cluster 7, which was primarily comprised of cells originating from samples treated with IL-1β (Fig S2), was highly enriched for Nos2 (iNOS) mRNA, and is mostly made up of β-cells, based on hormone gene expression (Fig 1C). Indeed, Cluster 7 is localized with the rest of the β-cell clusters in the UMAP projection (Fig 1B).

We used the Database for Annotation, Visualization and Integrated Discovery (DAVID) (Dennis et al, 2003) to identify significantly enriched categories among the genes up-regulated in Cluster 7 (Fig 1D). By far the most enriched category was "Immune response," with an enrichment score of nine. This group includes genes such as guanylate binding protein 5 (Gbp5), β defensin 1 (Defb1), CD40 antigen (Cd40), aconitate decarboxylase 1 (Acod1), Toll-like receptor 2 (Tlr2), an apolipoprotein B editing complex member (Apobec3), and a subunit of the transporter associated with antigen processing (Tap2) (Fig 1E). A number of guanylate binding proteins, a family of GTPases with antiviral activities (Anderson et al, 1999), were also strongly up-regulated in Cluster 7 cells (Fig 1F). Several genes encoding proteins with antiviral roles are also enriched in Cluster 7, including viperin (Rsad2); several IFN-induced genes that help recognize and degrade viral RNA (Ifit1, Ifit3, and Isg20); an IFN-stimulated gene that inhibits viral replication (Isg15); interferon regulatory factor 1 (Irf1), a master regulator of the antiviral defense; and a member of the 2′–5′-oligoadenylate synthetase family (Oasl2), which facilitates RNase L-mediated degradation of viral RNA (Fig 1G) (Schneider et al, 2014; Zhu et al, 2014). As expected, in addition to Nos2, we observed enrichment in several other genes involved in NF-κB signaling, including Icam1, encoding inflammatory intercellular adhesion molecule-1; Sod2, encoding manganese-dependent superoxide dismutase; Nfkbia, encoding an NF-κB inhibitor; and Nfkb1, encoding subunit 1 of NF-κB (Fig 1H). Finally, one of the most highly enriched genes in Cluster 7, Cxcl10, is a proinflammatory chemokine (Cardozo et al, 2003), and is up-regulated along with several other members of the chemokine family (Fig 1I).

### Distinct β-cell clusters are seen after 6-h cytokine exposure

Based on our initial clustering analysis (Fig 1B), there appeared to be several distinct groups of β-cells. To further characterize this heterogeneity, we performed a separate clustering analysis using only Ins1- and Ins2-expressing cells in our dataset (Clusters 0, 1, 4–8). Computationally isolating and clustering β-cells revealed 7 clusters (Figs 2A and S3A and Table S3). One (β-cell Cluster 4) appeared to only be present in cells treated with IL-1β, whereas another (β-cell Cluster 6) was only present in cells treated with

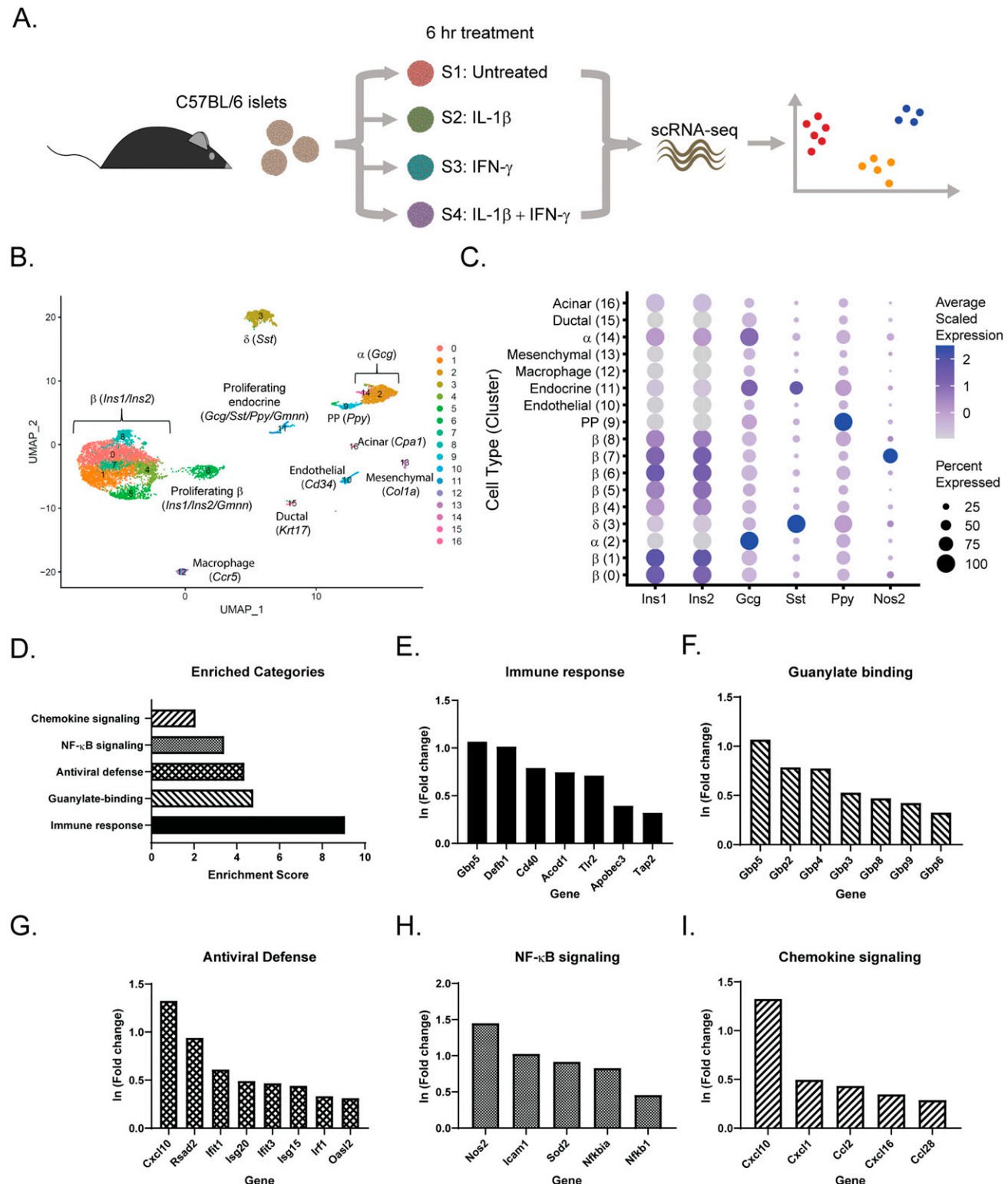

**Figure 1. Single-cell RNA sequencing of mouse islet cells after 6 h cytokine exposure.**
**(A)** Schematic of experimental design. **(B)** Uniform Manifold Approximation and Projection plot depicting clusters of cells from all four samples. Clusters are numbered and color-coded according to the legend shown at the right. Cell identity was assigned based on enrichment for genes indicated. **(C)** Dot plot indicating expression levels and percent of cells expressing *Ins1*, *Ins2*, *Gcg*, *Sst*, *Ppy*, and *Nos2* mRNA in each of the 17 clusters. **(D)** Significantly enriched genes in cluster 7 were subjected to functional annotation clustering using the Database for Annotation, Visualization and Integrated Discovery (DAVID). Select enriched categories of genes are shown. **(E, F, G, H, I)** Selected representative genes from the categories shown in (D). Fold change refers to changes in Cluster 7 compared with all other clusters.

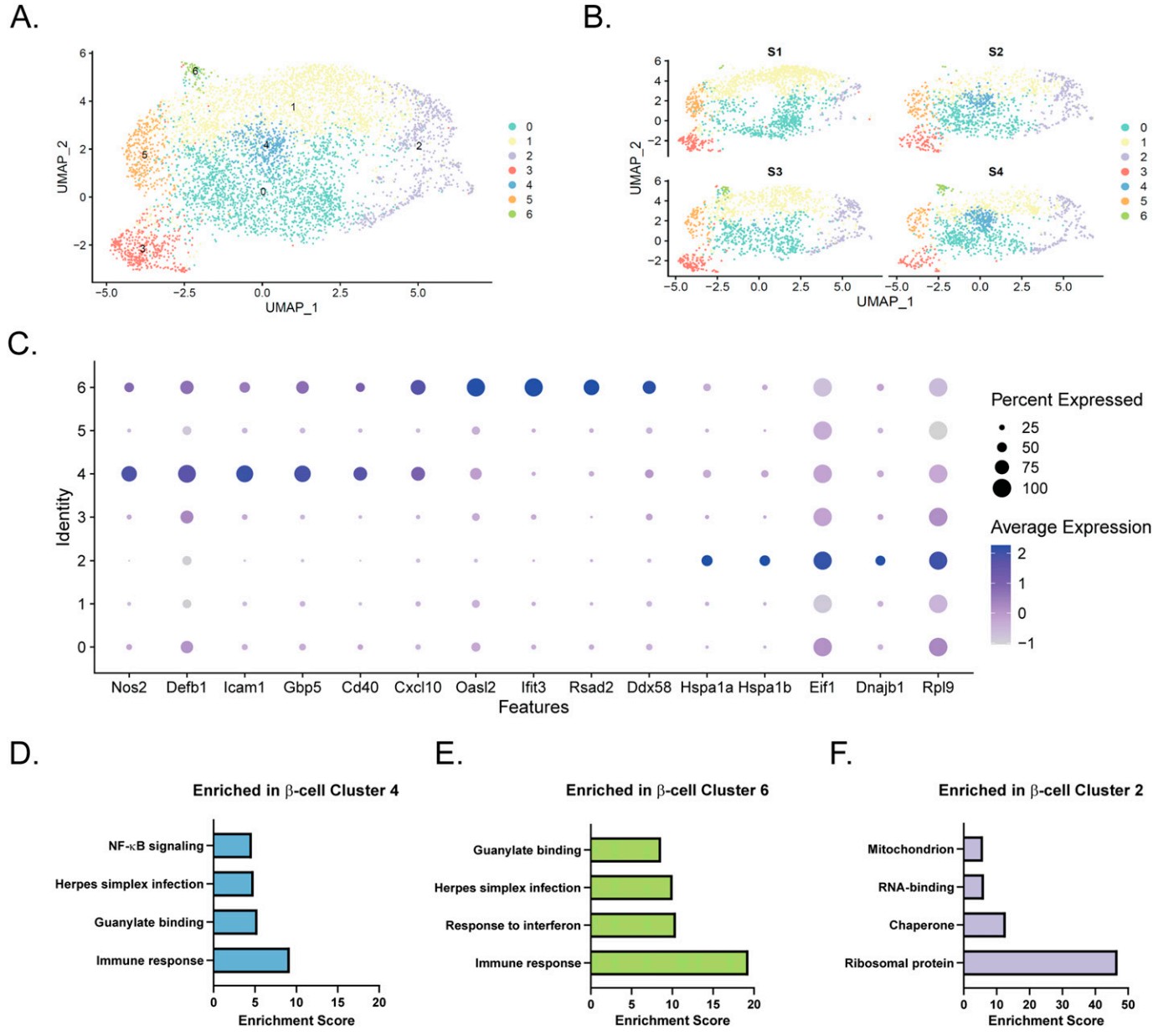

**Figure 2. β-cell clusters after 6-h cytokine exposure.**
**(A)** Uniform Manifold Approximation and Projection plot depicting clusters of β-cells from all four samples. Clusters are numbered and color-coded according to the legend shown at the right. **(B)** Uniform Manifold Approximation and Projection plots segregated by sample. **(C)** Dot plot indicating expression levels and percent of cells expressing selected genes in each of the seven β-cell clusters. **(D, E, F)** Significantly enriched genes in cluster 4 (D), cluster 6 (E), or cluster 2 (F) were subjected to functional annotation clustering using DAVID. Select enriched categories of genes are shown for each cluster.

IFN-γ (Fig 2B), and both clusters had high expression of *Nos2* mRNA (Fig 2C). To determine the distinguishing characteristics of these two clusters of iNOS-expressing β-cells, we again used DAVID to identify enriched categories of up-regulated genes. There were several categories enriched in both clusters (Immune response, Guanylate binding, and Herpes simplex infection), but β-cell Cluster 6, the smaller of the two, had a higher correlation to these categories (Fig 2D and E). Closer examination revealed that IL-1β-regulated genes, including *Defb1, Icam1, Gbp5,* and *Cd40* were primarily expressed in β-cell Cluster 4, while IFN-induced genes, including

*Cxcl10, Oasl2, Ifit3, Rsad2* (viperin), and *Ddx58* (Rig-I) were enriched in β-cell Cluster 6 (Fig 2C). Differential expression analysis between these two clusters corroborates these conclusions (Fig S3B and C), as several of the top 10 genes increased in Cluster 4 compared to Cluster 6 are related to IL-1 signaling (*Nfkb1* and *Icam1*), whereas genes increased in Cluster 6 include those regulated by IFNs (*Rsad2, Ifit3,* and *Oasl2*). In addition to the *Nos2*-expressing clusters, we also identified a cluster (β-cell Cluster 2) enriched for ribosomal proteins, chaperones, RNA-binding proteins, and mitochondrial proteins (Fig 2F). Genes enriched in these cells included those

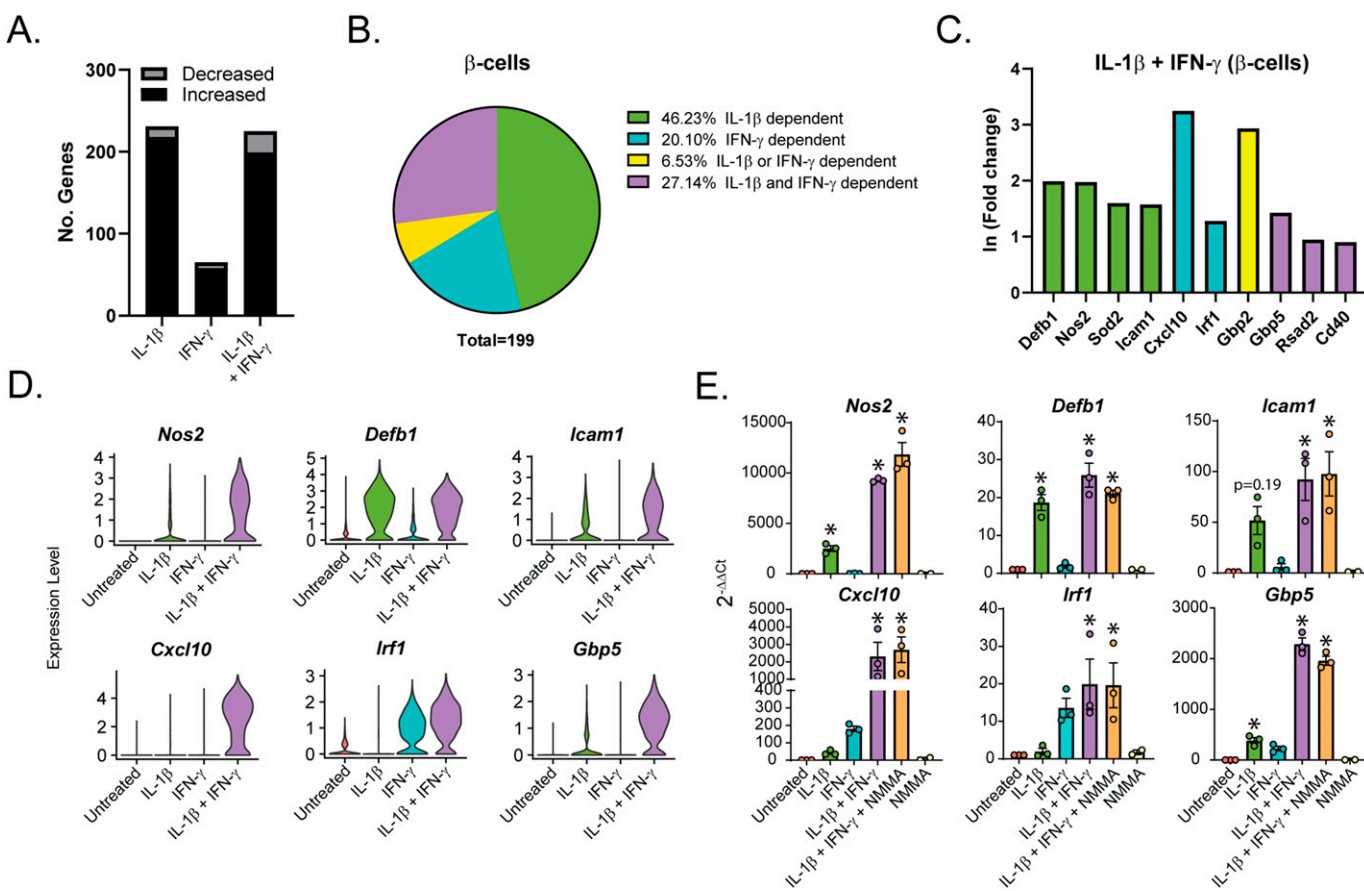

**Figure 3. Cytokine requirements for different genes in β-cells.**
**(A)** Stacked bar plot showing the number of significantly up-regulated or down-regulated genes in β-cells in S2 (IL-1β), S3 (IFN-γ), or S4 (IL-1β + IFN-γ) compared with S1 (Untreated). ln (Fold change) > 0.5 and an adjusted *P*-value < 0.01 were used as cutoffs to determine significance. **(B)** Genes significantly up-regulated in β-cells treated with IL-1β + IFN-γ (S4) were compared with genes significantly up-regulated in β-cells treated with IL-1β (S2) or with IFN-γ (S3). Pie chart shows the percentage of up-regulated genes in S4 that are IL-1β–dependent (also up-regulated in S2), IFN-γ–dependent (also up-regulated in S3), IL-1β- or IFN-γ–dependent (up-regulated in S2, S3, and S4), or IL-1β– and IFN-γ–dependent (only up-regulated in S4). **(C)** ln (fold change) from single-cell RNA sequencing data of selected representative genes in the categories shown in (B). Green bars: IL-1β–dependent. Teal bars: IFN-γ–dependent. Yellow bars: IL-1β- or IFN-γ–dependent. Purple bars: IL-1β and IFN-γ–dependent. **(D)** Violin plots showing the expression level of selected genes in β-cells from each of the four samples. **(E)** Expression of selected genes (compared with *Gapdh* levels) determined by qRT-PCR using mouse islets treated with the indicated treatments for 6 h. Concentrations are as follows: 10 U/ml IL-1β, 150 U/ml IFN-γ, and 2 mM NMMA. Error bars represent SEM. qRT-PCR results are from three independent experiments with statistical significance indicated. *P* < 0.05 (versus untreated).

encoding subunits of heat shock protein 70 (*Hspa1a* and *Hspa1b*), eukaryotic translation initiation factor 1 (*Eif1*), heat shock protein 40 (*Dnajb1*), and ribosomal protein L9 (*Rpl9*) (Fig 2C). Enrichment of genes involved in these stress responses suggests that cells in β-cell Cluster 2 had been stressed before treatment.

### Cytokine requirements for differential gene expression in β-cells

To determine how the cytokines individually and together influenced gene expression, we performed differential expression analysis comparing the β-cells in Samples 2, 3, and 4 to those in Sample 1 (Table S4). We used a threshold of 0.5 ln(fold change) and an adjusted *P*-value of 0.01 to determine if expression of a gene was significantly changed. In response to: (1) IL-1β alone, 231 genes (218 up, 13 down); (2) IFN-γ alone, 65 genes (58 up, 7 down); and (3) both cytokines, 225 genes (199 up, 26 down) were modified (Fig 3A). Of the genes up-regulated in response to both IL-1β and IFN-γ (Sample 4),

92 (46.23%) were IL-1β–dependent (also up-regulated in Sample 2), 40 (20.10%) were IFN-γ–dependent (also up-regulated in Sample 3), 13 (6.53%) were induced by either cytokine (up-regulated in Samples 2, 3, and 4), and 54 (27.14%) required both cytokines for significant induction (they were only up-regulated in Sample 4) (Fig 3B). Examples of up-regulated genes from scRNA-seq data are shown (Fig 3C and D), and selected genes were validated by qRT-PCR using mouse islets (Fig 3E). *Nos2* and *Sod2* (manganese-dependent superoxide dismutase) were among the IL-1β–dependent genes, as expected, as were *Defb1* and *Icam1* (Fig 3C–E). Among the IFN-γ–dependent genes we found *Irf1*, as anticipated, as well as *Cxcl10* (Fig 3C–E). The latter gene was significantly induced in response to IFN-γ by scRNA-seq, but, interestingly, was about 13-fold higher in response to both cytokines together (Fig 3C and E), suggesting that IL-1β dramatically augments its expression. Because of this dramatic increase in expression with exposure to both cytokines, the increase in expression by IFN-γ alone did not reach

statistical significance by qRT-PCR (Fig 3E). Very few genes were induced in all three cytokine-treated samples, but an antiviral guanylate-binding protein (*Gbp2*) was among them (Fig 3C). Finally, among the genes requiring both cytokines for significant induction were *Gbp5*, *Rsad2*, and *Cd40* (Fig 3C–E). *Gbp5* was not significantly up-regulated by IL-1β alone via scRNA-seq analysis, but was by qRT-PCR (Fig 3D and E). As expected, these genes are not regulated by nitric oxide, as addition of NOS inhibitor $N^{G}$-mono-methyl-L-arginine (NMMA) does not prevent the up-regulation by cytokines (Fig 3E). We were surprised to observe induction of antiviral genes by either IL-1β (*Defb1*, *Gbp2*, and *Gbp5*) or IFN-γ (*Irf1* and *Gbp2*), as antiviral function is typically attributed to IFN signaling, but not to IL-1 signaling (Samuel, 2001). We did not observe induction of pro-apoptotic genes in any of our samples, as has been suggested (Fig S4) (Grunnet et al, 2009). Together, these results suggest that (1) in β-cells, IL-1β has a greater influence on gene expression than IFN-γ, but a number of genes require both cytokines to be significantly induced; and (2) IL-1β and IFN-γ both individually up-regulate antiviral genes in β-cells.

### *Nos2* mRNA is increased in a subset of δ- and α-cells

Our initial clustering analysis revealed that most "cytokine-responsive" cells (those that express *Nos2* mRNA in response to IL-1β and IFN-γ exposure) were found in Cluster 7 of our overall dataset, a β-cell–enriched cluster (Fig 1C). Indeed, comparison of the expression of *Nos2* and *Ins1* in Sample 4 showed enhanced accumulation of *Nos2* mRNA in 71% of β-cells (Fig 4A and B). However, closer examination of the α- and δ-cell clusters (Clusters 2 and 3, respectively) revealed that a subset of these non–β-cells also express *Nos2* mRNA in response to treatment with both cytokines (Fig 4C and E). 69% of δ-cells and 39% of α-cells express *Nos2* in Sample 4, suggesting that subsets of these cell types are "cytokine responsive" (Fig 4D and F and Tables S5 and S6). *Nos2* mRNA was up-regulated in β-cells and δ-cells in response to IL-1β alone or in combination with IFN-γ (Fig 4G and H), whereas α-cells required both cytokines (Fig 4I). In agreement, mouse αTC1 cells increase *Nos2* mRNA in response to 6 h exposure to 10 U/ml IL-1β and 150 U/ml IFN-γ (Fig S5A). Fifty percent of PP-cells expressed *Nos2* in response to both cytokines, but this increase was not statistically significant (Fig S6C and Table S7). These findings are unexpected given that β-cells are thought to be the only islet endocrine cell type capable of generating nitric oxide upon IL-1β stimulation (Corbett et al, 1992b; Corbett & McDaniel, 1995; Arnush et al, 1998).

### IL-1β and IFN-γ induce many of the same genes in β-, δ-, and α-cells

Because β-, δ-, and α-cells all respond to cytokine stimulation by expressing iNOS, we hypothesized that they may display similar changes in gene expression in addition to *Nos2*. To test this, we compared the expression of the top up-regulated genes in β-cells to their expression levels in δ- and α-cells. These genes were significantly induced in all three cell types, with comparable fold changes (Fig 5A and Tables S4–S6). In addition to fold change, we also compared the percentage of each cell type with detectable transcripts for each gene. For most genes, fewer than 5% of β-cells expressed the transcript in untreated cells, but this increased to at least 70% in response to both cytokines (Fig 5B). For a few genes, however, including *Mt2* and *Sod2*, a similar percentage of cells

expressed the transcript in both untreated and cytokine-treated samples (Fig 5B). Importantly, the patterns of gene expression were comparable in both δ- and α-cells (Fig 5C and D), suggesting that β-cells and non–β endocrine cells respond to initial cytokine exposure in a similar manner.

### Cytokine-responsive genes in non–β-endocrine cells

Like our analysis of the β-cells (Fig 3), we performed differential expression analysis comparing the δ-, α-, and PP-cells in Samples 2, 3, and 4 to those in Sample 1 (Tables S5–S7). In contrast to the β-cells, in which 46% of genes were induced by IL-1β, (Fig 3A), only 25% and 32% of genes were induced by IL-1β in δ- and α-cells, respectively (Fig 6A and C). Accordingly, δ- and α-cells required stimulation by both cytokines to induce a greater percentage of cytokine-dependent genes than did β-cells (Fig 6A and C), suggesting that β-cells may be more sensitive to IL-1β stimulation than other islet endocrine cells. In support of this conclusion, 35% and 38% of the genes that were significantly induced by IL-1β alone in β-cells required both IL-1β and IFN-γ for induction in δ- and α-cells, respectively. *Defb1*, *Sod2*, and *Icam1* are induced in response to IL-1β, whereas *Irf1* is dependent on IFN-γ in all three cell types (Figs 3C–E and 6B and D–F). There are some genes, such as *Cxcl10*, which are induced by IFN-γ alone in β-cells but required stimulation by both cytokines in δ- and α-cells (Figs 3C–E and 6B and D–F). In addition, *Cd40* and *Rsad2* are induced by both cytokines in β- and δ-cells but are not significantly induced in α-cells (Figs 3C–E and 6B and D–F). In accordance with our scRNA-seq data, αTC1 cells treated with IL-1β and IFN-γ for 6 h have increased mRNA accumulation of *Icam1*, *Cxcl10*, and *Gbp5* in a nitric oxide–independent manner (Fig S5C–E). Although not detected by scRNA-seq, *Ptgs2* mRNA levels (encoding cyclooxygenase 2 or Cox2) were increased by 6-h IL-1β exposure in αTC1 cells, further suggesting the induction of NF-κB driven gene expression in non-β endocrine cells in response to cytokines (Fig S5B). In general, the expression of fewer genes was increased in PP-cells in response to IL-1β when compared to the other islet endocrine cells, but *Cxcl10*, *Icam1*, *Defb1*, and *Gbp5* expression levels were increased after stimulation by both cytokines (Fig S6A–C).

These data suggest that although many of the same genes are induced in all islet endocrine cells in response to cytokines, β-cells may be more sensitive to stimulation by individual cytokines than are δ-, α-, and PP-cells.

### Repression of endocrine cell identity genes in an IL-1β–dependent manner

Others have reported repression of β-cell identity factors, including *Pdx1*, *Mafa*, and *Slc2a2*, in response to prolonged IL-1β treatment (of at least 24 h) (Cardozo et al, 2001a; Nordmann et al, 2017). Therefore, we were surprised to observe repression of these and other β-cell identity factors (*Ucn3*, *Nkx6-1*, and *Hadh*) after only 6 h of treatment (Fig 7A and E). Repression is IL-1β–dependent, as it was observed only in Samples 2 (IL-1β alone) and 4 (IL-1β + IFN-γ), but not in Sample 3 (IFN-γ alone) (Fig 7A and E). *Pdx1*, *Mafa*, and *Nkx6-1* are transcription factors that maintain β-cell identity and promote insulin transcription and secretion (Matsuoka et al, 2004; Henseleit et al, 2005; Gao et al, 2014); *Ucn3* encodes a maturation factor (Blum et al, 2012); *Slc2a2* encodes the glucose transporter Glut2 that has a glucose sensing function in β-cells (Guillam et al, 1997); and *Hadh* encodes an enzyme that participates in β-oxidation of fatty acids (Hardy et al, 2007). Repression of identity factors did not coincide

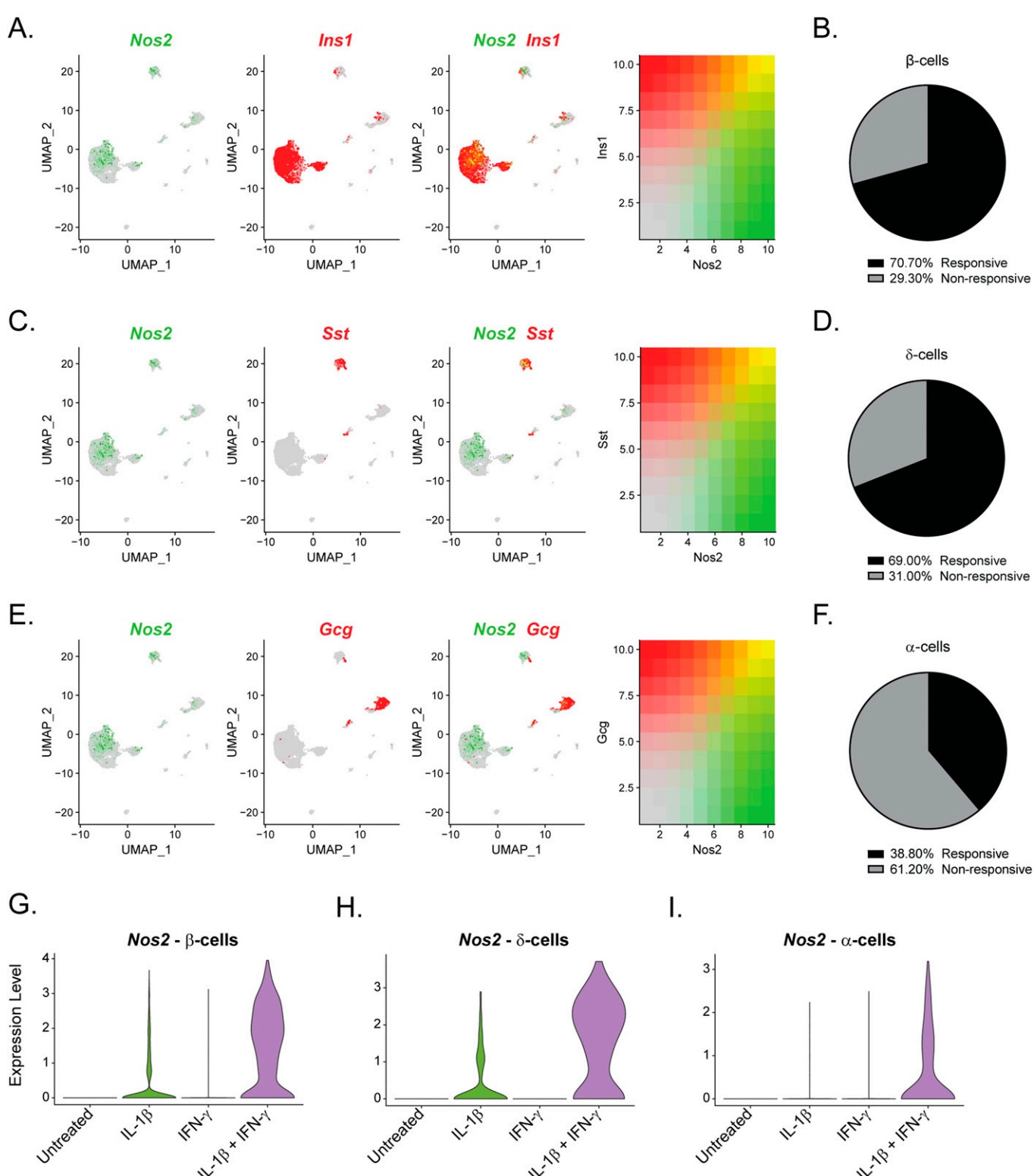

**Figure 4. Populations of α- and δ-cells express *Nos2* mRNA in response to IL-1β.**
**(A, C, E)** Uniform Manifold Approximation and Projection plots showing the overlap in expression of *Nos2*, shown in green, and *Ins1* (A), *Sst* (C), or *Gcg* (E) shown in red. Intermediary colors denote cells that co-express *Nos2* with each respective hormone. Expression levels of each gene are indicated to the right of each panel. **(B, D, F)** Pie charts showing the percentage of cytokine-responsive (*Nos2*-expressing) and nonresponsive (*Nos2* non-expressing) β-cells (B), δ-cells (D), or α-cells (F) in Sample 4. **(G, H, I)** Violin plots showing the expression levels of *Nos2* mRNA in β-cells (G), δ-cells (H), or α-cells (I) in each sample.

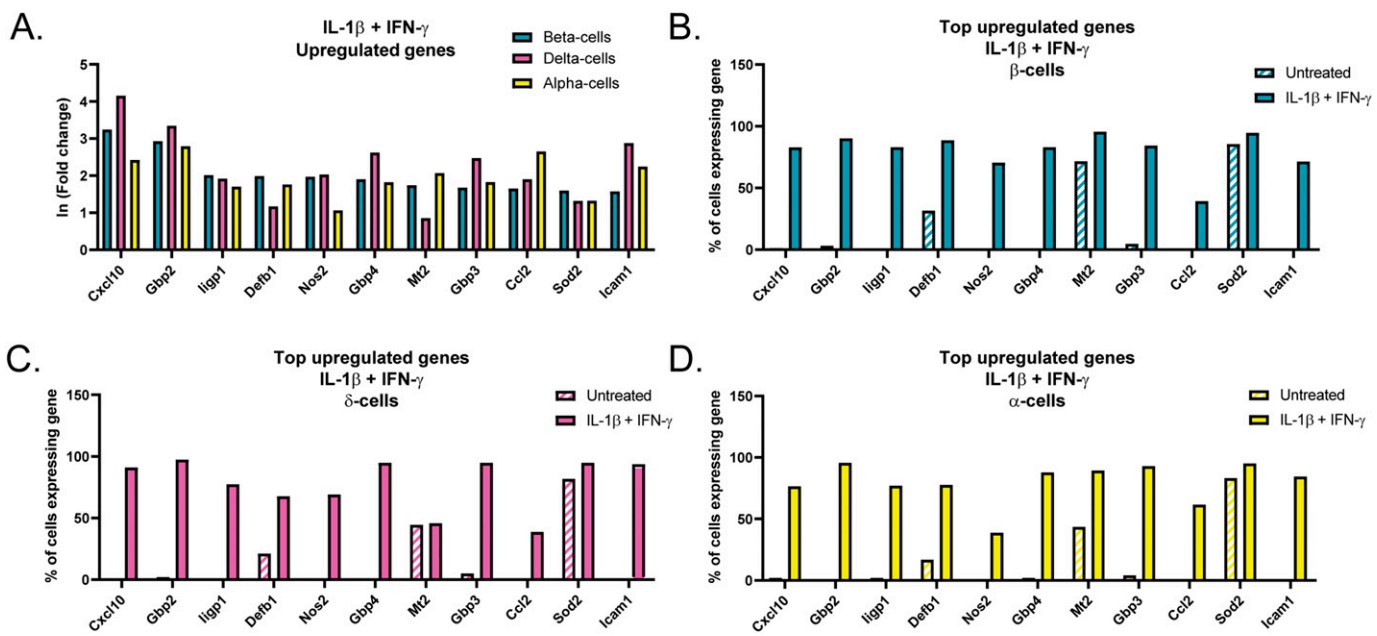

**Figure 5. Top up-regulated genes in endocrine cells in response to IL-1β and IFN-γ.**
**(A)** Histogram showing ln (fold change) of selected genes in β-cells (turquoise bars), δ-cells (magenta bars), and α-cells (yellow bars) from Sample 4 compared with Sample 1. The top 11 genes induced in β-cells are shown. **(B, C, D)** Histograms showing the percent of β-cells (B), δ-cells (C), and α-cells (D) in the untreated sample (hatched bars) and in the sample treated with IL-1β + IFN-γ (solid bars) expressing transcripts of selected genes shown in (A).

with increased expression of β-cell disallowed genes, such as *Ldha* (lactate dehydrogenase A), *Slc16a1* (monocarboxylate transporter 1), and *Hk1* (hexokinase) (Fig S7A) nor with genes associated with dedifferentiation, such as *Gast* (gastrin), *Neurog3*, and *Aldh1a3* (Fig S7B) (Talchai et al, 2012; Kim-Muller et al, 2016; Dahan et al, 2017). Furthermore, we observed repression of δ-cell identity factor *Hhex*, which promotes δ-cell specification and somatostatin transcription, in an IL-1β–dependent manner (Zhang et al, 2014) (Fig 7B and E). Repression of α-cell identity factor *Arx*, which promotes α-cell specification and maturity (Wilcox et al, 2013), did not achieve statistical significance, but is observed when evaluated by both scRNA-seq and qRT-PCR (Fig 7C and E). Not only was the expression level of each factor reduced, but the percentage of cells with detectable levels of the gene was also dramatically reduced by cytokines (Fig 7D). Importantly, repression of these factors was not affected by addition of NMMA (Fig 7E). These data suggest that IL-1β represses not only β-cell identity factors, but also δ- and α-cell identity factors in a nitric oxide–independent manner.

### Islet stress is negatively correlated with "cytokine responsiveness"

Whereas most β-cells treated with IL-1β and IFN-γ (Sample 4) expressed *Nos2* (71%), nearly one third (29%) failed to increase *Nos2* expression. We and others have shown that islet stress such as heat shock or ER stress attenuates the ability of cytokines to stimulate iNOS expression in β-cells (Bellmann et al, 1995; Scarim et al, 1998; Burkart et al, 2000; Weber et al, 2004b). Islet stress, as identified by increased expression of heat shock protein (Hsp)70, is associated with inhibition of signaling such that IL-1–induced NF-κB activation and MAPK signaling, and IFN-γ–stimulated STAT phosphorylation,

are attenuated (Bellmann et al, 1995, Scarim et al, 1998; Weber et al, 2003, 2004a, 2004b). Indeed, comparison of the expression patterns of *Nos2* and *Hspa1a*, encoding a subunit of Hsp70, suggested a negative correlation between the two genes in β-cell clusters: cells expressing high levels of *Hspa1a* do not express *Nos2* in response to IL-1β and IFN-γ (Figs 8A and B and S8A). These cytokine nonresponsive cells appeared to correlate by UMAP visualization to β-cell Cluster 2, which was enriched in genes encoding ribosomal proteins and chaperones (Fig 2F). Indeed, differential expression analysis using *Nos2*-expressing versus non-expressing β-cells in Sample 4 revealed significant enrichment of genes encoding heat shock proteins including *Hspa1a*, *Hspa1b*, *Dnajc24*, and *Dnajc3* (Fig 8E and Table S8).

In addition to the negative correlation between *Nos2* and heat shock gene expression, cells expressing high levels of *Hspa1a* also did not express antiviral genes, including *Gbp5*, *Gbp2*, *Gbp4*, *Defb1*, *Rsad2*, and *Irf1* in response to cytokine treatment (Figs 8C and F and S8A and B). Other cytokine-induced genes follow a similar pattern of expression in cells expressing high levels of *Hspa1a*, including *Icam1* and *Cxcl10* (Fig S8A, C, and D). Finally, β-cells with no or low expression of *Hspa1a* had reduced expression of identity gene *Hadh* in response to IL-1β treatment, whereas cells with high expression of *Hspa1a* maintained *Hadh* expression (Fig 8A and D). A similar pattern of expression was observed between *Hspa1a* and *Mafa* (Fig S8A and E). In accordance, differential expression analysis revealed that cells failing to increase *Nos2* had significantly higher expression of identity genes, including *Mafa*, *Hadh*, *Ucn3*, *Slc2a2*, and *Pdx1* in response to cytokine treatment (Fig 8E). These observations suggest that activation of the heat shock response (or stress response) not only prevents cytokine-induced *Nos2* expression,

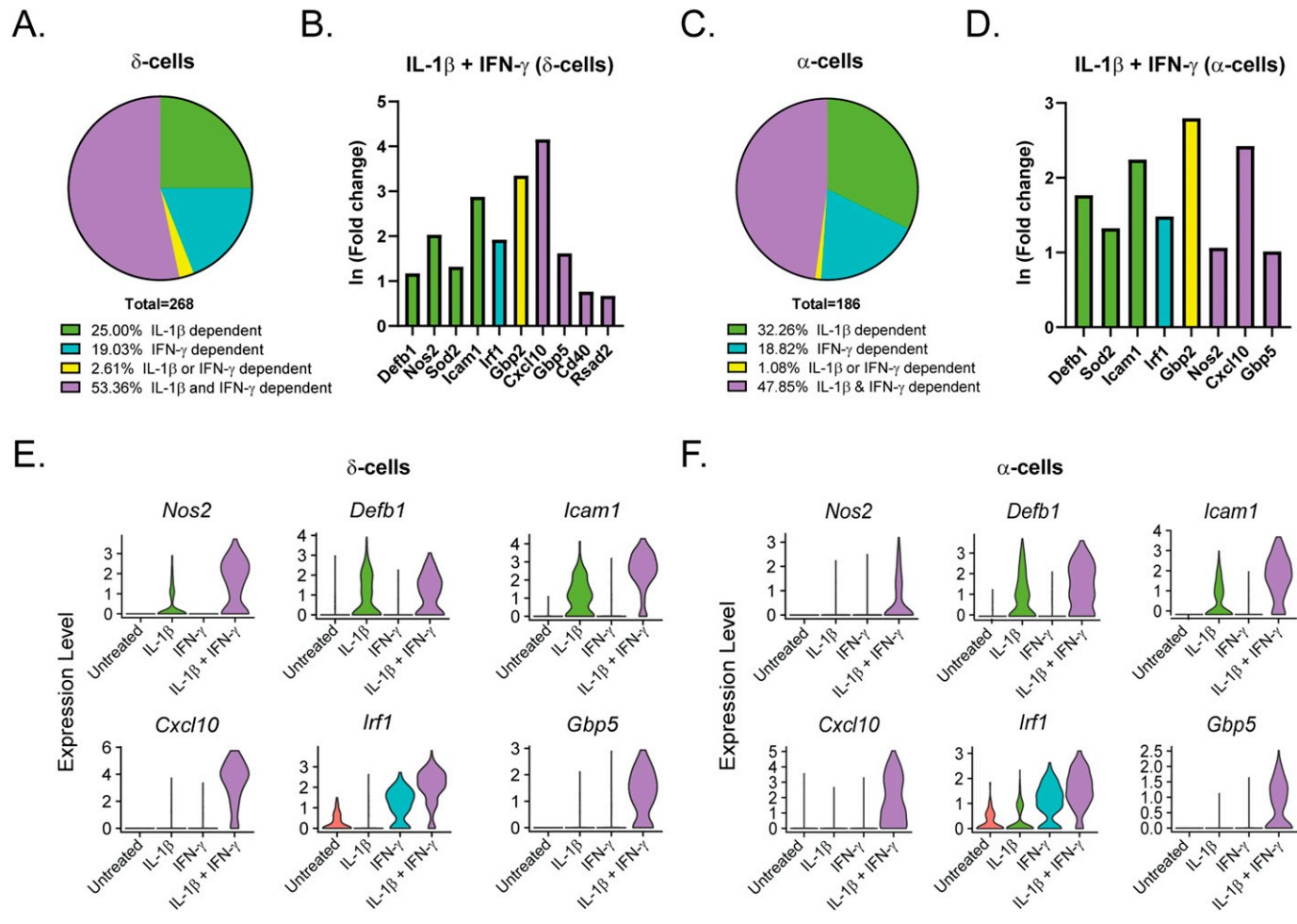

**Figure 6. Cytokine-responsive genes in δ- and α-cells.**
**(A, C)** Genes significantly up-regulated in δ- and α-cells treated with IL-1β + IFN-γ (S4) were compared with genes significantly up-regulated in δ- and α-cells with IL-1β (S2) or with IFN-γ (S3). Pie charts show the percentage of up-regulated genes in S4 δ-cells (A) and α-cells (C) that are IL-1β–dependent (also up-regulated in S2), IFN-γ–dependent (also up-regulated in S3), IL-1β– or IFN-γ–dependent (up-regulated in S2, S3, and S4), or IL-1β– and IFN-γ–dependent (only up-regulated in S4). **(B, D)** ln (fold change) of selected representative genes in the categories shown in (A) and (C) in S4 δ-cells (B) and S4 α-cells (D). Green bars: IL-1β–dependent. Teal bars: IFN-γ–dependent. Yellow bars: IL-1β– or IFN-γ–dependent. Purple bars: IL-1β– and IFN-γ–dependent. **(E, F)** Violin plots showing the expression levels of selected genes in δ-cells (E) and α-cells (F) from each of the four samples.

but also prevents other aspects of cytokine signaling, including induction of antiviral genes and identity gene repression.

# Discussion

Autoimmune, or type 1 diabetes, is characterized by selective destruction of insulin-producing pancreatic β-cells that is preceded by an inflammatory reaction in and around the islets of Langerhans (Gepts, 1965; Like et al, 1985; Wicker et al, 1986; Bergman & Haskins, 1997). T lymphocytes and macrophages that infiltrate the islet participate in this inflammatory reaction, releasing cytokines such as IFN-γ and IL-1β (Wright et al, 1988; Wright & Lacy, 1989). IL-1β is known to inhibit β-cell function and cause islet damage in a nitric oxide–dependent manner (Mandrup-Poulsen et al, 1985; Bendtzen et al, 1986). Nitric oxide is produced by β-cells after IL-1–induced

iNOS expression and attenuates insulin secretion by inhibiting oxidative metabolism and decreasing ATP levels in β-cells (Corbett et al, 1993; Corbett & McDaniel, 1995).

Over the years, studies have partially elucidated how cytokines and nitric oxide affect islet or β-cell gene expression (Cardozo et al, 2001a, 2001b; Hughes et al, 2011; Meares et al, 2013). However, these have been limited by the use of intact islets, which represents a heterogeneous cell population, or purified β-cells, which ignores the potential influences of population heterogeneity. In addition, few studies have explored the effects of cytokines on islet non-β-cells. Finally, many previous studies were performed after prolonged treatment periods (24 h or greater), which miss immediate or early effects of cytokines and may be confounded by secondary and tertiary changes due to the actions of the cytokines. Here, we chose a short treatment period of 6 h, or an early treatment, that allowed us to determine how islet cells initially respond to cytokine stimulation, and diminish the effects of secondary responses, such

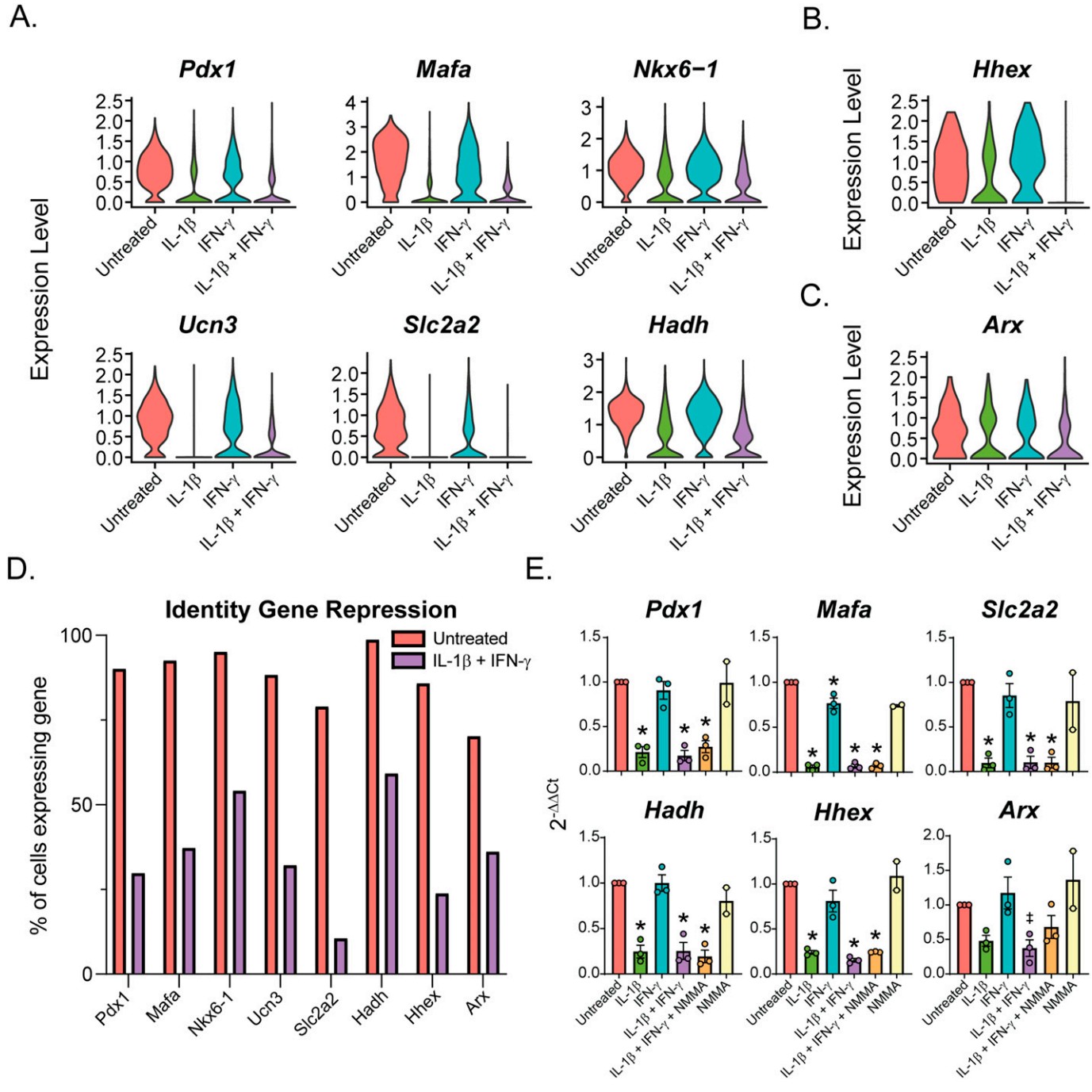

**Figure 7. IL-1β represses endocrine cell identity genes.**
**(A, B, C)** Violin plots showing the expression level of selected identity genes in β-cells (A), δ-cells (B), and α-cells (C) from each of the four samples. **(D)** Histogram showing the percent of untreated cells and cells treated with IL-1β + IFN-γ expressing detectable transcripts of selected genes. **(E)** Expression of selected genes (compared with *Gapdh* levels) determined by qRT-PCR using mouse islets treated with the indicated treatments for 6 h. Concentrations are as follows: 10 U/ml IL-1β, 150 U/ml IFN-γ, and 2 mM NMMA. Error bars represent SEM. qRT-PCR results are from three independent experiments with statistical significance indicated. *P < 0.05, ‡P = 0.05 (versus untreated).

as nitric oxide, that may modify gene expression profiles (Corbett et al, 1992a).

The initial clustering analysis of our scRNA-seq data revealed 17 distinct clusters, delineating several different cell types, most of which were β-cells (Figs 1 and S1). The primary *Nos2*-expressing

cluster, Cluster 7, was enriched for genes involved in the immune response, guanylate binding, antiviral defense, NF-κB signaling, and chemokine signaling, and primarily consisted of β-cells (Fig 1). When we computationally isolated and re-clustered only the β-cells (Clusters 0, 1, 4–8), two clusters were enriched for *Nos2* (Figs

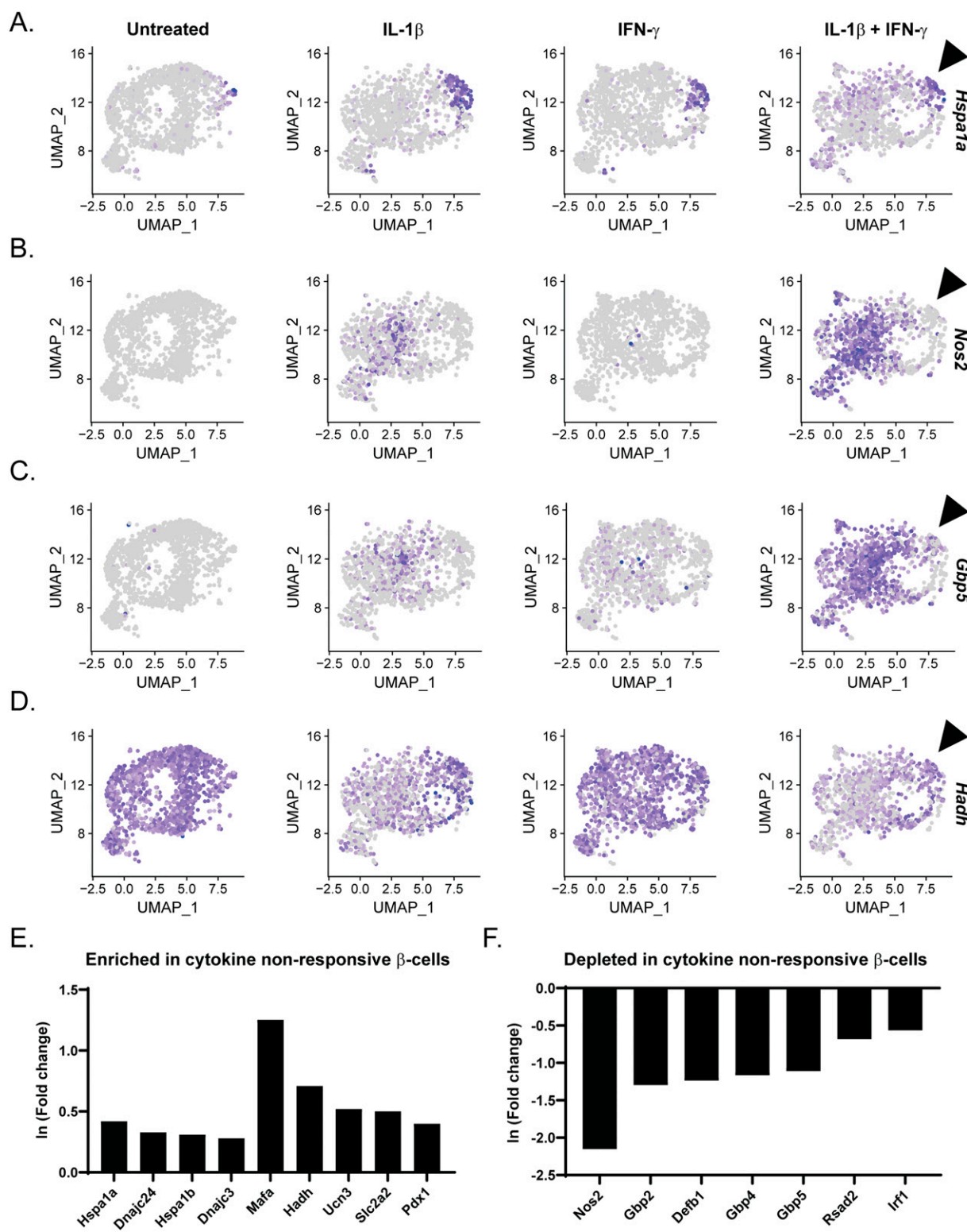

**Figure 8. Heat shock protein gene expression is negatively correlated with *Nos2* expression and identity gene repression.**
**(A, B, C, D)** Uniform Manifold Approximation and Projection plots segregated by sample and colored to indicate expression level of *Hspa1a* (A), *Nos2* (B), *Gbp5* (C), and *Hadh* (D) in β-cell clusters. Arrowhead points out the population of cells with high expression of *Hspa1a* that fails to induce *Nos2* and *Gbp5* or to repress *Hadh* in response to IL-1β and IFN-γ. Panel (A) is repeated in Fig S8A. **(E, F)** Differential expression analysis comparing cytokine-responsive (*Nos2*-expressing) and nonresponsive (*Nos2* non-expressing) β-cells treated with IL-1β and IFN-γ. Histograms show the expression level (ln (fold change)) of selected genes that are significantly up-regulated (E) or down-regulated (F) in cytokine nonresponsive β-cells.

2 and S3). One (β-cell Cluster 4) appeared to be primarily enriched for IL-1β–induced genes, whereas the other (β-cell Cluster 6) was enriched for IFN-γ–induced genes. Intriguingly, we also identified a cluster of β-cells with high expression of ribosomal proteins, heat shock proteins, and other chaperones. This cluster persisted across the samples, suggesting that these cells were stressed before cytokine exposure.

Although not obvious in our initial clustering analysis, we found that subpopulations of δ- and α-cells express *Nos2* mRNA (Fig 4) in response to cytokines. Like β-cells, δ-cells increased *Nos2* in response to IL-1β alone, but α-cells required both IL-1β and IFN-γ to significantly increase *Nos2*. These findings were unexpected, given that previous studies using rodent and human islets have suggested that β-cells are the primary endocrine cellular source of iNOS-derived nitric oxide in the islet (Corbett et al, 1992b; Corbett & McDaniel, 1995; Arnush et al, 1998). There are many potential reasons for this discrepancy that include species-dependent effects (rat or human versus mouse) and differences in cytokines used (IL-1β alone versus IL-1β and IFN-γ). However, the most likely reason for the differences is that most colocalization studies did not specifically focus on iNOS expression in α- or δ-cells but focused on iNOS expression in insulin-positive cells. Thus, iNOS expression in non–β-endocrine cells was likely missed or attributed to non-endocrine cells (macrophages or endothelial cells) known to express iNOS and to be present in islets. With these caveats in mind, our studies indicate that all endocrine cell types in islets are responsive to cytokines, including enhanced expression of iNOS. Whether or not iNOS induction results in nitric oxide production in these non–β-cells, and how nitric oxide affects δ- and α-cell function and survival in the context of cytokine exposure has yet to be determined.

The similarities in response to cytokines among β-, δ-, and α-cells do not end with *Nos2* induction. Our analysis revealed that the top up-regulated genes in β-cells in response to IL-1β and IFN-γ (Sample 4) are similarly induced in δ- and α-cells, both in fold change and in percentage of cells expressing the transcript (Fig 5). In fact, of the top 100 genes induced in Sample 4 β-cells, 61 are also induced in δ-cells, and 66 are induced in α-cells. Similarities in the expression of genes that were repressed by cytokines were also observed. IL-1β has previously been shown to repress genes critical for maintaining β-cell function and identity, including *Pdx1*, *Mafa*, and *Slc2a2* (Cardozo et al, 2001a; Nordmann et al, 2017). However, these studies observe this repression after prolonged cytokine exposure of 24 h or longer (Cardozo et al, 2001a; Nordmann et al, 2017). We show a significant repression of these same genes after only 6 h of IL-1β treatment, suggesting that this is an immediate response to cytokine exposure rather than an indication of dedifferentiation after prolonged exposure (Fig 7). Furthermore, we found repression of genes that maintain δ- and α-cell identity, such as the transcription factors *Hhex* and *Arx*, in the δ- and α-cell populations, respectively (Fig 7). Like the repression of β-cell identity genes, repression of *Hhex* and *Arx* only occurs upon exposure to IL-1β, but not to IFN-γ alone. These exciting results suggest that: (1) IL-1β is a primary mediator of the decrease in expression of genes associated with endocrine cell identity; (2) this action is not modified by IFN-γ; and (3) the loss of identity is not unique to β-cells but occurs in all islet endocrine cells.

Although IFN-γ is a known activator of antiviral responses (Samuel, 2001), our analysis revealed that IL-1β induces several protective genes in islet endocrine cells, a role for this cytokine that is not well-characterized. *Defb1* (β-defensin 1) is an antimicrobial peptide (Semple & Dorin, 2012) and is robustly increased in β-, δ-, and α-cells in a IL-1β–dependent manner (Figs 3 and 6). Expression of this gene was confirmed by qRT-PCR analysis and is nitric oxide independent (Fig 3E). Guanylate-binding proteins are known to be IFN-regulated and have broad antiviral and antimicrobial activities (Anderson et al, 1999), but here, we found *Gbp2* to be regulated by either IL-1β or IFN-γ alone (Figs 3 and 6). In addition, *Gbp5* was significantly induced by IL-1β in islets by qRT-PCR analysis and was trending to be induced by IL-1β by scRNA-seq analysis in β-cells (Fig 3), but not in δ- or α-cells (Fig 6). Recently, both *Gbp2* and *Gbp5* have been shown to inhibit viral glycoprotein maturation (Braun et al, 2019). These findings demonstrate that both IL-1β and IFN-γ individually induce antiviral genes in islet endocrine cells.

There is a difference between β-cells and non–β-endocrine cells when comparing the changes in gene expression in response to the individual cytokines. In β-cells, IL-1β had the greatest effect on gene expression, significantly increasing more than 200 genes, whereas IFN-γ significantly increased about 65 genes (Fig 3). This trend held true when assessing the genes induced in cells treated with both cytokines, in which about 46% were IL-1β–dependent and 20% were IFN-γ–dependent, whereas 27% were only induced under conditions in which both cytokines were present (Fig 3). δ-, α-, and PP-cells had a greater percentage of genes (53%, 48%, and 69%, respectively) that required the presence of both cytokines for significant up-regulation (Figs 6 and S6). The difference in cytokine requirement may suggest that endocrine non–β-cells are less sensitive to the effects of either cytokine alone, or may be due to limitations in our experimental approach, as the α-, δ-, and PP-cell populations were at least five times smaller than the β-cell population captured (5,436 cells versus 921, 651, and 159 cells). Because of lower cell numbers, statistical significance was more difficult to achieve in α-, δ-, and PP-cells. Independent of the individual cytokine requirements, our results show that all islet endocrine cells respond to cytokine exposure in a similar manner: by rapidly up-regulating iNOS, antiviral genes, and additional genes involved in the immune response, and by repressing transcription factors and genes critical for maintenance of cell identity.

Our analyses suggest that all islet endocrine cells respond to cytokines in a similar manner, including up-regulation of *Nos2* and antiviral pathways and repression of islet identity genes (Fig 9). Generally, cytokines and nitric oxide are viewed as damaging to β-cells based on observations that exogenous IL-1β treatment impairs oxidative phosphorylation and insulin secretion and eventually causes β-cell death with prolonged exposure in a nitric oxide–dependent manner (Mandrup-Poulsen et al, 1985; Bendtzen et al, 1986; Southern et al, 1990; Welsh et al, 1991; Corbett & McDaniel, 1992, 1995; Corbett et al, 1992a, 1993; Cunningham & Green, 1994; Mandrup-Poulsen, 1996). However, our recent studies suggest that the ability of nitric oxide to inhibit mitochondrial oxidation and ATP generation protects β-cells from apoptosis (Oleson et al, 2016) and viral infection (Stafford et al, 2020). Our observations here that IL-1β induces antiviral genes (Fig 3) and fails to induce apoptotic mediators (Fig S4) dovetail with these recent findings and further support the idea that the primary functions of IL-1β and nitric oxide are to protect β-cells from damage.

Finally, we observed that cytokine responsiveness in β-, α-, and δ-cells was heterogeneous. For example, *Nos2* was induced in 71% of β-cells, 69%

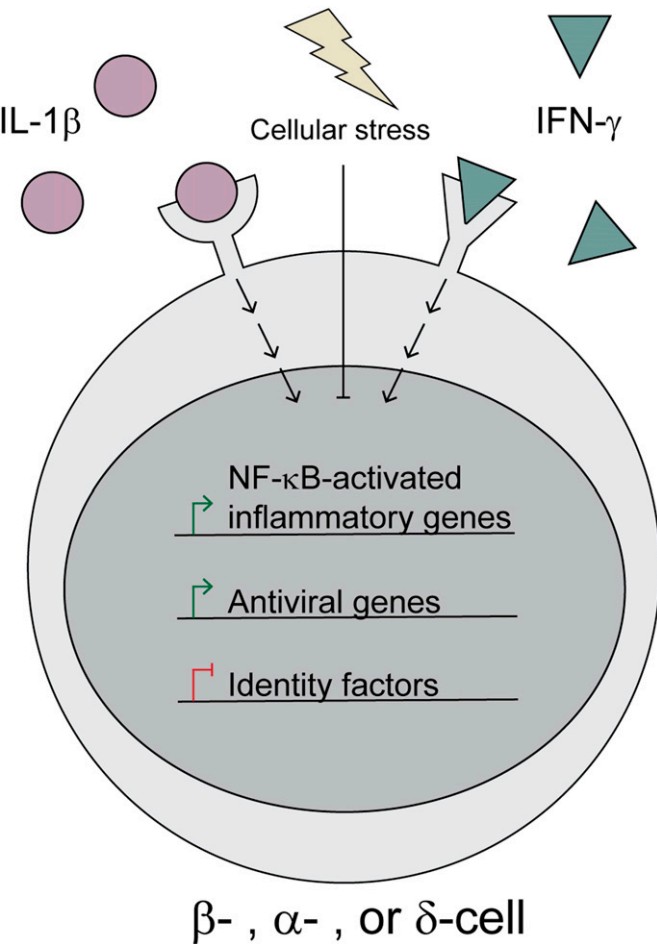

**Figure 9.  Model.**
Upon stimulation by IL-1β and IFN-γ, islet endocrine cells (β-, α-, and δ-cells) respond by up-regulating inducible nitric oxide synthase (*Nos2*) and a subset of antiviral genes, while simultaneously down-regulating identity factors, such as *Pdx1*, *Mafa*, *Hhex*, and *Arx*. However, induction of cellular stress, such as heat shock or ER stress, attenuates these aspects of cytokine signaling.

of δ-cells, and 39% of α-cells in response to IL-1β and IFN-γ (Fig 4). The population of cells that failed to express *Nos2* under these conditions corresponded with β-cell Cluster 2. The identifying feature of this cluster was elevated expression of genes encoding heat shock proteins and chaperones (Figs 2 and 8). Indeed, we observed that cells with high *Hspa1a* expression failed to express *Nos2* (Fig 8), or in other words, the expression of heat shock genes is enhanced in endocrine cells that fail to respond to cytokines. This finding is in agreement with previous studies demonstrating that in response to heat shock (Scarim et al, 1998) or ER stress (Weber et al, 2004a) cytokines fail to stimulate iNOS expression in rodent and human islets (Steer et al, 2006). Furthermore, this stress response is typical of human islet preparations (Welsh et al, 1995) and is likely responsible for the conclusions that nitric oxide does not mediate the damaging effects of cytokines on human islets (Eizirik et al, 1994; Burkart et al, 2000). More interesting are the findings that cells with high expression of *Hspa1a* also maintained expression of identity genes and did not express antiviral genes after cytokine treatment, suggesting a negative correlation between heat shock and cytokine signaling more broadly than just iNOS expression (Figs 8 and S8). It is likely

that the stress signature observed in this cluster is an artifact of the islet isolation process and does not represent an endogenous cell population. Together with the observation that cytokine-induced nitric oxide promotes heat shock protein expression in rat islets (Helqvist et al, 1991), our results suggest that the heat shock response may provide an internal "off switch" to prevent cytokine-mediated islet damage after protective signaling has been initiated. In other words, nitric oxide may not only protect β-cells through inhibition of mitochondrial oxidation (Oleson et al, 2016; Stafford et al, 2020), but may also provide a means to shut down prolonged cytokine signaling, which is known to be damaging.

# Materials and Methods

## Materials, cells, and animals

Male C57BL6/J mice were purchased from Jackson Laboratories and housed in the MCW Biomedical Resource Center. All animal use and experimental procedures were approved by the Institutional Animal Care and Use Committees at the Medical College of Wisconsin. αTC1 Clone 6 cells (CRL-2934) were purchased from American Type Culture Collection. Connaught Medical Research Laboratories 1066 medium, DMEM, HBSS, ι-glutamine, sodium pyruvate, Hepes, penicillin, and streptomycin were purchased from Thermo Fisher Scientific. FBS was purchased from HyClone. Trypsin (0.05% in 0.53 mM EDTA) was purchased from Corning. Human recombinant IL-1β and mouse IFN-γ were purchased from PeproTech. N$^{G}$-Monomethyl-ʟ-arginine (NMMA) is from Enzo Life Sciences.

## Islet isolation, islet and cell culture, and treatment

Islets were isolated from adult male C57BL6/J mice by collagenase digestion and were cultured at 37°C and 5% $CO_2$ in Connaught Medical Research Laboratories supplemented with 10% heat-inactivated FBS and containing 5.5 mM glucose as previously described (Kelly et al, 2003). Islets from eight mice were pooled before separation into four samples for cytokine treatment. αTC1 Clone 6 cells were maintained at 37°C and 5% $CO_2$, and were cultured in DMEM supplemented with 10% heat-inactivated FBS and containing 16.7 mM glucose, as previously described (Waas et al, 2020). Intact islets or adherent αTC1 cells were untreated or were treated for 6 h with 10 U/ml IL-1β alone, 150 U/ml IFN-γ alone, the combination of the two cytokines with or without 2 mM NMMA, or NMMA alone before dissociation and preparation for sequencing or before RNA isolation for qRT-PCR.

## scRNA-seq

After cytokine treatment, islets were dissociated into single-cell suspensions by incubation in 0.48 mM EDTA in phosphate buffered saline followed by agitation in 1 mg/ml trypsin in $Ca^{2+}/Mg^{2+}$-free HBSS. Cells were filtered and resuspended in HBSS + 0.04% BSA. Single-cell suspensions were loaded into the Chromium Controller (10x Genomics). scRNA-seq libraries were prepared using the Chromium Single Cell 3′ v3 Reagent Kit (10x Genomics) according to manufacturer's protocol.

Libraries were loaded onto a NextSeq 500/550 High Output Kit v2.5 flow cell (150 cycles; Illumina) with the following conditions: 26 cycles for read 1, 98 cycles for read 2, and 8 cycles for the i7 index read. CellRanger (10x Genomics) functions "mkfastq" and "count" were used to demultiplex the sequencing data and generate gene-barcode matrices. Sequencing depth was as follows: Sample 1 (108,744,355), Sample 2 (42,013,628), Sample 3 (70,207,661), and Sample 4 (54,588,606). All scRNA-seq analysis was performed in R (version 3.6.1) using the package Seurat (version 3.1.0) (Butler et al, 2018). Number of genes detected per cell and percent of mitochondrial genes were plotted, and outlier cells were removed (number of genes less than 200 or greater than 4,000, or percent mitochondrial genes over 12%) to filter out doublets and cells with low read quality, leaving 7,750 of the original 12,571 cells. Cell cycle genes were regressed. Principal component analysis was performed, and the top 30 principal components were used for UMAP analysis, with clustering performed using the Louvain algorithm. All samples were normalized using Seurat's default normalization settings. Briefly, reads in each cell for each gene were divided by the total number of reads within that cell, multiplied by a factor of 10,000, and transformed using the natural logarithm. Fold change values are expressed as natural log (ln) throughout the article, as this is the default method used by Seurat.

### qRT-PCR

Total RNA was isolated from mouse islets or $\alpha$TC1 cells using the RNeasy kit (QIAGEN). First-strand synthesis was performed using oligo(dT)s and Thermo Fisher Scientific Maxima H Minus reverse transcriptase per the manufacturer's instructions. Quantitative PCR was performed using the SsoFast EvaGreen supermix (Bio-Rad) and a Bio-Rad CFX96 Real-Time system. Primers were purchased from Integrated DNA Technologies and are listed in Table S9. For relative quantification, gene expression was normalized to *Gapdh* using the comparative ΔCt method (Nolan et al, 2006).

### Statistical analysis

For differential expression analysis, *P*-values were calculated using the Wilcoxon test and adjusted to avoid false positives using Bonferroni correction. The threshold level of significance was set to adjusted *P*-value < 0.01 and ln(fold-change) > 0.5. For qRT-PCR analysis, statistical comparisons were made between groups using one-way ANOVA with Tukey post hoc test in GraphPad Prism software, and the threshold level of significance was set to *P* < 0.05.

## Data Availability

Sequencing data from this publication have been deposited in the National Center for Biotechnology Information (NCBI) Gene Expression Omnibus database under accession number GSE156175.

## Supplementary Information

## Acknowledgements

The authors thank Dr. Polly Hansen and Joshua Stafford (Department of Biochemistry, Medical College of Wisconsin, Milwaukee, WI) for technical assistance and Dr. Polly Hansen, Aaron Naatz, Joshua Stafford, and Chay Teng Yeo (Department of Biochemistry, Medical College of Wisconsin, Milwaukee, WI) for helpful discussions related to this project and for proofreading the manuscript. This research was completed in part with computational resources and technical support provided by the Research Computing Center at MCW. Funding: This work was supported by the National Institute of Diabetes and Digestive and Kidney Diseases grant DK-052194 (to JA Corbett), the National Institute of Allergy and Infectious Diseases grant AI-044458 (to JA Corbett) and AI-125741 and AI-148403 (to W Cui), the Juvenile Diabetes Research Foundation grant SRA-2019-829-S-B (to JA Corbett), and a grant from the Medical College of Wisconsin Cancer Center (to JA Corbett). JS Stancill was supported by the National Heart, Lung, and Blood grant T32-HL134643. MY Kasmani was supported by the National Institute of Diabetes and Digestive and Kidney Diseases grant DK-127526 and is a member of the Medical Scientist Training Program at the Medical College of Wisconsin, which is partially supported by a training grant from the National Institute of General Medical Sciences (T32-GM080202). This work was also supported by gifts from the Scott Tilton Foundation and from the Forest County Potawatomi Foundation.

### Author Contributions

JS Stancill: conceptualization, data curation, formal analysis, supervision, validation, investigation, visualization, project administration, and writing—original draft, review, and editing.
MY Kasmani: software, formal analysis, visualization, methodology, and writing—original draft, review, and editing.
A Khatun: data curation, investigation, methodology, and writing—review and editing.
W Cui: resources, supervision, funding acquisition, methodology, project administration, and writing—original draft, review, and editing.
JA Corbett: conceptualization, resources, supervision, funding acquisition, project administration, and writing—original draft, review, and editing.

### Conflict of Interest Statement

The authors declare that they have no conflict of interest.

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
