## [Reviewer comments · Life Science Alliance]

Life Science Alliance

Single-cell RNA-sequencing of mouse islets exposed to proinflammatory cytokines

Jennifer Stancill, Moujtaba Kasmani, Achia Khatun, Weiguo Cui, and John Corbett

DOI: <https://doi.org/10.26508/lsa.202000949>

Corresponding author(s): John Corbett, Medical College of Wisconsin and Jennifer Stancill, Medical College of Wisconsin

Review Timeline:

Submission Date:	2020-10-26
Editorial Decision:	2021-01-05
Revision Received:	2021-02-03
Editorial Decision:	2021-03-27
Revision Received:	2021-04-01
Accepted:	2021-04-07

Scientific Editor: Shachi Bhatt

Transaction Report:

January 5, 2021

Re: Life Science Alliance manuscript #LSA-2020-00949-T

Prof. John A. Corbett
Medical College of Wisconsin
Biochemistry
8701 Watertown Plank Rd
Milwaukee, WI 53226

Dear Dr. Corbett,

Thank you for submitting your manuscript entitled "Single-cell RNA-sequencing of mouse islets exposed to proinflammatory cytokines" to Life Science Alliance. The manuscript was assessed by expert reviewers, whose comments are appended to this letter.

As you will note from the reviewers' comments, your manuscript was received by both the referees. While reviewer 1 only has minor requests, reviewer 2 does have a number of technical concerns which should be addressed prior to further consideration of the manuscript at LSA. We would thus like to invite you to resubmit a revised version of this manuscript that addresses all of the reviewers' concerns.

Thank you for this interesting contribution to Life Science Alliance. We are looking forward to receiving your revised manuscript.

Sincerely,

Shachi Bhatt, Ph.D.
Executive Editor
Life Science Alliance
<https://www.lsjournal.org/>
Tweet @SciBhatt @LSAJournal

- A letter addressing the reviewers' comments point by point.
- An editable version of the final text (.DOC or .DOCX) is needed for copyediting (no PDFs).
- High-resolution figure, supplementary figure and video files uploaded as individual files: See our detailed guidelines for preparing your production-ready images, <https://www.life-science-alliance.org/authors>
- Summary blurb (enter in submission system): A short text summarizing in a single sentence the study (max. 200 characters including spaces). This text is used in conjunction with the titles of papers, hence should be informative and complementary to the title and running title. It should describe the context and significance of the findings for a general readership; it should be written in the present tense and refer to the work in the third person. Author names should not be mentioned.

B. MANUSCRIPT ORGANIZATION AND FORMATTING:

Reviewer #1 (Comments to the Authors (Required)):

Stancill et al present single cell RNAseq data and analysis on mouse islet treated with a short (6 hr) time with cytokines interleukin 1beta, interferon gamma or both. These cytokines are thought to be involved in the destruction of pancreatic beta cells in Type 1 diabetes, and their effects on islets have been studied extensively. Two aspects of this study add value: first, the analysis of single islet cells and second, using only a 6 hr treatment with cytokine so the direct effect was seen rather than one later down a cascade of events. Mouse islets were treated, dispersed and analyzed, with over 7700 single islet cells passing quality control. The most important findings were supported and

included :1) there was heterogeneity of response to the cytokines with similar genes were induced in the responsive cells whether beta cells (70% of total beta cells), glucagon producing alpha cells (39%) or somatostatin expressing delta cells (69%); 2) IL1beta had the most effect on the beta cell identity; 3) IL1beta had antiviral effects which had previously been attributed only to IFNg; 4) the combination of both cytokines had greater effects than the sum of both alone.; 5) there was no immediate induction of apoptotic genes;) the beta cells were more sensitive to either single cytokine than the other islet cells.

The authors provide an excellent introduction and explanation for the rationale for doing scRNAseq on islets treated with cytokines, providing references to original resources rather than falling into the unfortunate habit of referring only to recent reviews. While there are always issues whether the heterogeneity found in sc RNA seq is due to the inherent differences due to cells out of synchrony or the less extensive array of expressed genes detected, the differences here were robust and often confirmed with PCR.

One suggestion that would improve this paper is to include supplemental tables of the differentially expressed genes. It would be particularly of interest to the islet biologists to know what were the 13 genes (6 identified in text) repressed in beta cells by Il1b and the 7 by IFNg. Were there additional ones from the combination?

Reviewer #2 (Comments to the Authors (Required)):

Stancill et al describe gene expression changes in mouse islets upon exposure to proinflammatory cytokines followed single-cell RNA-seq. Authors identify Nos2 expressing cells and other gene expression changes, some of these changes were further confirmed using qPCR. Although this study attempts to provide understanding of role of cytokines in gene expression changes in islets, however, there are several caveats that should be addressed.

1) How many samples per group were used? What is the batch of treatments, 10x genomics emulsion formation and sequencing? What are sequencing depth for each of the samples? What are the median genes and UMIs in each sample and in each of the clusters shown in Figure 1B. How were the cytokine doses and treatment duration determined?

2) 10x genomics emulsion formation is prone to doublet cell capture (~5%). Authors should elaborate on how the doublets were identified and removed.

3) What are the gene expression differences in samples within the treatment group? The Cluster 7 in Figure 1B has been highlighted over and over again. This cluster is formed of few cells and doesn't separate well from the other groups of cells (cluster 0, 1, 4 etc.) and therefore it could simply be a result of not applying optimal clustering parameters. If the resolution is reduced, it will merge with neighboring clusters. Single-cell datasets are prone to sample processing (tissue dissociation) effects that sometimes induce stress related genes. Cluster 5 shows clear stress gene signature based on Figure S1 (authors made no effort to indicate what color indicates which cell type in the heatmap). Any cell type of interest (in this case Cluster 7, Nos2 positive cells) should be represented in multiple samples and should not be accounted by one sample. Authors have not made any efforts to clarify this issue.

4) Although authors indicate that Ins1 and Ins2 are markers of beta cells, Figure 1C. shows these genes expressed in almost all cell types. This indicates ambient RNA contamination that arises during the tissue dissociation processes. Processing multiple samples instead of one will help determine the background gene expression is indeed due to ambient RNA (technical) or biological

observation. In addition, it will help deduce the extent of such ambient RNA contamination to other cell types.

5) Cluster 6 and cluster 11 are indicated as proliferating cells. However, these cell clusters lack classic and abundantly expressed markers such as Mki67 and Top2a (Figure S1). More analysis on these cell cluster identification is required.

6) To identify differentially expressed genes among the different treatment groups, authors should use averaged gene expression (normalized to 10k, TP10k) profiles for a given cell type and then use DeSeq2 or similar statistical tools. Pathway analysis tools like DAVID can be used to aid the observations from DeSeq2 differential gene expression analysis. In figure 1 (E-I), authors do not make it clear what do the fold changes indicate. Are these cluster 7 verses all other clusters?

7) In figure 2, authors attempt to subcluster the beta cell types. The clusters are scattered, again indicating that optimal parameters were not used. Authors should provide a differential gene expression heatmap indicating top 10 differentially expressed genes in each of the 7 clusters. The Nos2 expressing b-cells are now shown to group into two clusters (cluster 4 and cluster 6). Are these two clusters split of cluster 7 in Figure 1B?

8) To determine differentially expressed genes between Nos2 positive cluster 4 and cluster 6, authors should use Seurat parameter (ex. "FindMarkers") and indicate top 10-20 genes.

9) Authors mention: "In addition, to the Nos2-expressing clusters, we also identified a cluster (b-cell Cluster 2) enriched for ribosomal proteins, chaperones, RNA-binding proteins, and mitochondrial proteins (Fig. 2F)." As mentioned earlier, b-cell cluster 2 expresses stress response genes which are often seen in single cell datasets and are result of tissue dissociation. (Brink et al, PMID: 28960196).

10) Authors consistently mention that "cluster 7" in Figure C are the "cytokine-responsive" cells since these cells express Nos2 in response to IL-1b and IFN-γ exposure. However many other clusters (cluster 0, 3, 8, 11, 14) show expression of Nos2. In fact, more than 25% of cells in cluster 0 and cluster 14 express Nos2.

11) To make it readable and easy to understand, authors should use cell type names in Figure 1C rather than using the cluster numbers.

Reviewer #1 (Comments to the Authors):

1) One suggestion that would improve this paper is to include supplemental tables of the differentially expressed genes. It would be particularly of interest to the islet biologists to know what were the 13 genes (6 identified in text) repressed in beta cells by IIIb and the 7 by IFN γ . Were there additional ones from the combination?

Response: We appreciate this suggestion and agree. We have included the suggested supplementary tables in the revised manuscript.

Reviewer #2 (Comments to the Authors):

1) How many samples per group were used? What is the batch of treatments, 10x genomics emulsion formation and sequencing? What are sequencing depth for each of the samples? What are the median genes and UMIs in each sample and in each of the clusters shown in Figure 1B. How were the cytokine doses and treatment duration determined?

Response: While single cell RNAseq was performed on one sample per treatment group, key findings were confirmed by qPCR for classes of genes increased and decreased in expression levels. Single cell capture was performed using the 10X Genomics Chromium Single Cell 3' v3 Reagent Kit. Sequencing was performed using the Illumina NextSeq 500/550 High Output Kit v2.5 flow cell (150 cycles). We have updated the text to include these requested details in the Methods. The sequencing depth of each Sample was added to the revised text, and we have added Supplemental Tables listing the mean genes and UMIs per cluster per sample. The results are also shown below for your convenience.

Mean genes per sample per cluster:

Cluster	Sample 1	Sample 2	Sample 3	Sample 4
0	3961	2750	3030	2973
1	3852	2514	2996	2837
2	3843	2307	2739	2715
3	3978	2357	2710	2560
4	3949	2736	3181	3084
5	3462	1774	2201	2244
6	3927	2655	3011	2971
7	3727	2581	3000	2822
8	3798	2557	2863	2634
9	3452	2058	2402	2380
10	3982	2107	2478	3042
11	4118	2765	2989	2635
12	3203	2018	2595	2619
13	3468	1992	2567	2307
14	4361	3494	3594	3652
15	3903	2480	3324	3003
16	3319	2251	1714	1974

Mean UMIs per sample per cluster:

Cluster	Sample 1	Sample 2	Sample 3	Sample 4
0	22111	9401	11476	12897
1	23036	8767	12361	13305
2	15941	7043	8216	9644
3	16318	6499	7786	8212
4	18980	7644	9899	9855
5	20685	5594	7861	9214
6	22556	9265	11285	12152
7	22762	9031	10995	13178
8	17303	7762	9182	9528
9	17245	6917	9075	10703
10	12911	4638	6380	9168
11	17562	8841	9161	9034
12	13053	5776	8470	9163
13	13433	5808	8422	7714
14	23908	12538	12770	15783
15	13971	6884	10627	9274
16	35727	10590	8757	9630

The cytokine doses were chosen on the basis of 30 yrs of experience working in this field and previous studies from our laboratory and others that have identified the concentration and time-dependent actions of cytokines

on iNOS expression in mouse β -cells (PMID 9153221, 10506184). The treatment duration of 6 hr was chosen because nitric oxide levels after this exposure are low and will not confound the gene expression results (PMID 1384465). We confirm the lack of a nitric oxide effect on gene expression in Fig 3, 7, and S5 using a NOS inhibitor (NMMA). Importantly the goals of our study were to determine early responses of islet cells to cytokine stimulation.

2) 10x genomics emulsion formation is prone to doublet cell capture (~5%). Authors should elaborate on how the doublets were identified and removed.

Response: Outlier cells were removed (number of genes less than 200 or greater than 4000). This has been clarified in the Methods.

3) What are the gene expression differences in samples within the treatment group? The Cluster 7 in Figure 1B has been highlighted over and over again. This cluster is formed of few cells and doesn't separate well from the other groups of cells (cluster 0, 1, 4 etc.) and therefore it could simply be a result of not applying optimal clustering parameters. If the resolution is reduced, it will merge with neighboring clusters. Single-cell datasets are prone to sample processing (tissue dissociation) effects that sometimes induce stress related genes. Cluster 5 shows clear stress gene signature based on Figure S1 (authors made no effort to indicate what color indicates which cell type in the heatmap). Any cell type of interest (in this case Cluster 7, Nos2 positive cells) should be represented in multiple samples and should not be accounted by one sample. Authors have not made any efforts to clarify this issue.

Response: Figures 3-6 focus on our differential expression analyses comparing gene expression differences across the treatments by cell type. We have now included new supplemental tables (Tables S4-S7) showing the genes that are differentially expressed based on those analyses. To address your comment about highlighting Cluster 7 - we have now included a supplemental figure (Fig S2) indicating the percentage of each cluster that originates from each of the 4 samples. We focused on Cluster 7 for much of the manuscript because this is the cluster of cells that is enriched for Nos2, which is characteristic of cytokine exposure. This new figure we included indicates that Cluster 7 is primarily made up of cells from the samples treated with IL-1 β (Samples 2 and 4). This is expected, since Nos2 expression is IL-1 β -dependent in β -cells. We would expect only minimal contribution to this cluster from untreated cells or cells treated only with IFN- γ . To address your other point, we have now re-labeled the heatmap in Fig S1 to indicate the cell types represented by each cluster.

4) Although authors indicate that Ins1 and Ins2 are markers of beta cells, Figure 1C. shows these genes expressed in almost all cell types. This indicates ambient RNA contamination that arises during the tissue dissociation processes. Processing multiple samples instead of one will help determine the background gene expression is indeed due to ambient RNA (technical) or biological observation. In addition, it will help deduce the extent of such ambient RNA contamination to other cell types.

Response: Because endocrine hormones are expressed at extremely high levels in islet cells, it is not surprising to observe ambient RNA contamination. Indeed, others have comparable RNA contamination using similar methodologies. In PMID 33432158 (*Scientific Reports* 2021) Figure S1, INS expression is found in human α -cells and GCG expression in human β -cells). In PMID 32302527 (*Cell Metabolism* 2020) Figure S2, INS expression is observed in human α -, δ -, and PP-cells. Furthermore, while we list the islet hormones as the identifier genes of the endocrine cell clusters in Fig 1C (as is customary in the field), the clusters can be further identified based on expression of other characteristic genes. A new supplementary table (Table S1) has been included which lists all genes enriched in each cluster. Therefore, while there may be low levels of RNA contamination of Ins1 and Ins2 in non- β -cell clusters, we are confident that the cell types have been identified correctly.

5) Cluster 6 and cluster 11 are indicated as proliferating cells. However, these cell clusters lack classic and abundantly expressed markers such as *Mki67* and *Top2a* (Figure S1). More analysis on these cell cluster identification is required.

Response: By our analysis, only 17 genes were enriched in Cluster 6. Other than Geminin (*Gmnn*), the remaining 16 genes are characteristic of β -cells and were similarly enriched in other β -cell clusters. Therefore, we have no other genes, besides *Gmnn*, to distinguish this cluster from the other β -cell clusters. A greater number of genes (200) were enriched in Cluster 11. If we consider the top 20 enriched genes, all, with the exception of *Gmnn*, are similarly enriched in at least one of the other non- β -endocrine clusters (2, 3, 9, or 14). In fact, most of the top 20 genes in Cluster 11 were enriched in multiple non- β -endocrine clusters. New Supplemental Table S1 is included to allow readers to see all genes enriched in these clusters. We were surprised to identify these clusters in our analysis, because islet endocrine cells are terminally-differentiated and are not proliferative. *Gmnn* is known to be expressed in replicative cells (PMID 12107111). Therefore, we have identified these clusters to the best of our ability as “proliferative” cells and have indicated that they are enriched for *Gmnn* (Fig 1B). We assessed *Mki67* and *Top2a* expression in all clusters and detected no expression (shown below). However, when we assessed cell cycle phase, Clusters 6 and 11 appear to consist entirely of cells in S Phase of the cell cycle (see below). Based on the expert reviewer’s concern, we tempered the identification of these clusters in the revised text, stating that *Gmnn* expression “suggest[s] that they may represent proliferative cells.”

6) To identify differentially expressed genes among the different treatment groups, authors should use averaged gene expression (normalized to 10k, TP10k) profiles for a given cell type and then use *DeSeq2* or similar statistical tools. Pathway analysis tools like *DAVID* can be used to aid the observations from *DeSeq2*

differential gene expression analysis. In figure 1 (E-I), authors do not make it clear what do the fold changes indicate. Are these cluster 7 verses all other clusters?

Response: We have updated the Methods to include more details about the normalization method: “All samples were normalized using Seurat's default normalization settings. Briefly, reads in each cell for each gene were divided by the total number of reads within that cell, multiplied by a factor of 10000, and transformed using the natural logarithm.” We have also updated the Legend of Figure 1 to clarify that the fold changes indicate changes in Cluster 7 compared to all other clusters.

7) In figure 2, authors attempt to subcluster the beta cell types. The clusters are scattered, again indicating that optimal parameters were not used. Authors should provide a differential gene expression heatmap indicating top 10 differentially expressed genes in each of the 7 clusters. The Nos2 expressing b-cells are now shown to group into two clusters (cluster 4 and cluster 6). Are these two clusters split of cluster 7 in Figure 1B?

Response: As requested by the reviewers, we have addressed each of the outlined issues in the new data presented in Fig S3, including a heat map of each of the Clusters of β -cells.

8) To determine differentially expressed genes between Nos2 positive cluster 4 and cluster 6, authors should use Seurat parameter (ex. "FindMarkers") and indicate top 10-20 genes.

Response: We performed the analysis you suggested and have included the top 10 genes increased in Cluster 4 vs Cluster 6 and vice versa in the revised Supplemental Figure S3.

9) Authors mention: "In addition, to the Nos2-expressing clusters, we also identified a cluster (b-cell Cluster 2) enriched for ribosomal proteins, chaperones, RNA-binding proteins, and mitochondrial proteins (Fig. 2F)." As mentioned earlier, b-cell cluster 2 expresses stress response genes which are often seen in single cell datasets and are result of tissue dissociation. (Brink et al, PMID: 28960196).

Response: Since the cells expressing stress response genes also do not respond to cytokines in the expected way, we think the stress occurred prior to the cytokine treatment, which, in our case, was prior to the islet dissociation for scRNA-seq. It is likely that the stress was induced by the islet isolation process, not by the dissociation to single cells. Regardless of the reason for the stress response genes, we now acknowledge in the Discussion that this is a result of the experimental process and does not likely represent an endogenous cell population. However, the inclusion of this population in our analysis allows us to speculate about why only ~75% of the β -cells treated with both cytokines responded by expressing Nos2 – likely because the remaining ~25% were stressed prior to the treatment. This conclusion aligns with previous studies (PMIDs: 9832444, 15315910, 7769124, 8706913). We think this is an important observation for the field because human islet preparations often express high levels of heat shock proteins, preventing them from responding properly to cytokine exposure (PMID: 10751413), and isolation stress is known to contribute to human islet cell death (PMID: 11141234). Moreover, this connection has been overlooked in the literature, and some studies have suggested that human islets do not respond to cytokines by expressing Nos2 (PMID: 7514190). Our data presented here supports the idea that human islets (as well as rodent islets) do not express Nos2 in response to cytokines when they are stressed.

10) Authors consistently mention that "cluster 7" in Figure C are the "cytokine-responsive" cells since these cells express Nos2 in response to IL-1b and IFN- γ exposure. However many other clusters (cluster 0, 3, 8, 11, 14) show expression of Nos2. In fact, more than 25% of cells in cluster 0 and cluster 14 express Nos2.

Response: We focused on Cluster 7 because the cells in this cluster are significantly enriched for Nos2 expression compared to expression in all other clusters. We were also interested in this cluster because it is primarily (if not entirely) made up of β -cells. The low percentage of β -cells in Cluster 0 that express Nos2, as you mention, is likely attributable to a small number of β -cells from Samples 2 and 4 (cells exposed to IL-1 β) being assigned to that cluster rather than Cluster 7. This is supported by the data presented in Fig S2. While the cell clustering is not perfect, we hope you will agree that the majority of β -cells expressing Nos2 reside in Cluster 7, which is why we chose to focus on those cells for much of the manuscript. Most of the clusters you mention as having cells expressing Nos2 are other cell types (non- β -cells), including Clusters 3, 8, 11, and 14. We would not expect those cells to be sufficiently different from others of the same cell type to cluster with a primarily β -cell cluster. As we discuss later in the manuscript (Fig 4), other islet cell types express Nos2 in response to cytokines, not just β -cells (including α -cells and δ -cells).

11) To make it readable and easy to understand, authors should use cell type names in Figure 1C rather than using the cluster numbers.

Response: We agree and have made this change.

March 27, 2021

RE: Life Science Alliance Manuscript #LSA-2020-00949-TR

Prof. John A. Corbett
Medical College of Wisconsin
Biochemistry
8701 Watertown Plank Rd
Milwaukee, WI 53226

Dear Dr. Corbett,

Thank you for submitting your revised manuscript entitled "Single-cell RNA-sequencing of mouse islets exposed to proinflammatory cytokines". We would be happy to publish your paper in Life Science Alliance pending final revisions necessary to meet our formatting guidelines.

While the reviewers were unable to comment on the revised manuscript, the revisions were assessed by our internal team of editors and an advisory member with the subject matter expertise and were deemed to have been appropriately performed.

Along with the points listed below, please also attend to the following,

- please add callouts for Figures S6A, B, C; S8A to your main manuscript text
- please use the [10 author names, et al.] format in your references (i.e. limit the author names to the first 10)
- Panel 8A and Panel S8A; Panel 8D first column Panel S8E first column are the same. Looks like that is by design, but we would appreciate it if you can clarify this in the figure legends, so the readers are aware that the figure panels have been repeated.

A. FINAL FILES:

-- High-resolution figure, supplementary figure and video files uploaded as individual files: See our

detailed guidelines for preparing your production-ready images, <https://www.life-science-alliance.org/authors>

B. MANUSCRIPT ORGANIZATION AND FORMATTING:

Thank you for your attention to these final processing requirements. Please revise and format the manuscript and upload materials within 5 days.

Sincerely,

Shachi Bhatt, Ph.D.
Executive Editor
Life Science Alliance
<https://www.lsjournal.org/>
Tweet @SciBhatt @LSAJournal

April 7, 2021

RE: Life Science Alliance Manuscript #LSA-2020-00949-TRR

Prof. John A. Corbett
Medical College of Wisconsin
Biochemistry
8701 Watertown Plank Rd
Milwaukee, WI 53226

Dear Dr. Corbett,

Thank you for submitting your Research Article entitled "Single-cell RNA-sequencing of mouse islets exposed to proinflammatory cytokines". It is a pleasure to let you know that your manuscript is now accepted for publication in Life Science Alliance. Congratulations on this interesting work.

DISTRIBUTION OF MATERIALS:

Again, congratulations on a very nice paper. I hope you found the review process to be constructive and are pleased with how the manuscript was handled editorially. We look forward to future exciting submissions from your lab.

Sincerely,

Shachi Bhatt, Ph.D.
Executive Editor
Life Science Alliance
<http://www.lsjournal.org>
Tweet @SciBhatt @LSAJournal